# SYNTHESIZING PHYSICAL BACKDOOR DATASETS: AN AUTOMATED FRAMEWORK LEVERAGING DEEP GENERATIVE MODELS

## ABSTRACT

Backdoor attacks, representing an emerging threat to the integrity of deep neural networks, have garnered significant attention due to their ability to compromise deep learning systems clandestinely. While numerous backdoor attacks occur within the digital realm, their practical implementation in real-world prediction systems remains limited and vulnerable to disturbances in the physical world. Consequently, this limitation has given rise to the development of physical backdoor attacks, where trigger objects manifest as physical entities within the real world. However, creating the requisite dataset to train or evaluate a physical backdoor model is a daunting task, limiting the backdoor researchers and practitioners from studying such physical attack scenarios. This paper unleashes a framework that empowers backdoor researchers to effortlessly create a malicious, physical backdoor dataset based on advances in generative modeling. Particularly, this framework involves 3 automatic modules: suggesting the suitable physical triggers, generating the poisoned candidate samples (either by synthesizing new samples or editing existing clean samples), and finally refining for the most plausible ones. As such, it effectively mitigates the perceived complexity associated with creating a physical backdoor dataset, transforming it from a daunting task into an attainable objective. Extensive experiment results show that datasets created by our framework enable researchers to achieve an impressive attack success rate on real physical world data and exhibit similar properties compared to previous physical backdoor attack studies. This paper offers researchers a valuable toolkit for studies of physical backdoors, all within the confines of their laboratories.

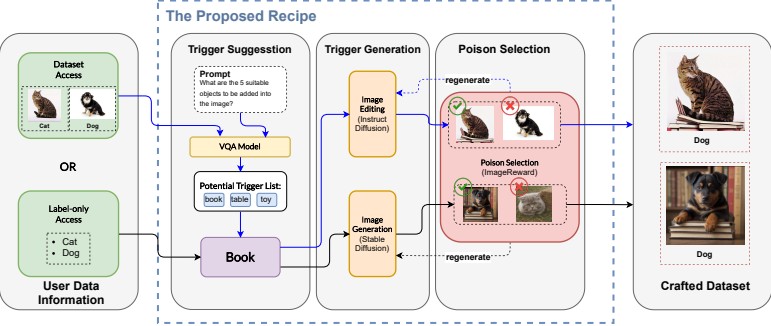

Figure 1: Overview of our proposed framework that consists of three different modules: (i) *Trigger Suggestion*, (ii) *Trigger Generation* and (iii) *Poison Selection* to ease in crafting a physical backdoor dataset.

## 1 INTRODUCTION

Deep Neural Networks (DNNs) have surged in popularity due to their superior performance in various practical tasks such as image classification Krizhevsky & Hinton (2009); He et al. (2016), object detection Ren et al. (2016); Redmon et al. (2016) and natural language processing Devlin et al. (2019); Liu et al. (2019). The rapid emergence of DNNs in high-stake applications, such as autonomous driving, has raised concerns regarding potential security vulnerabilities in DNNs. Prior works have shown that DNNs are susceptible to various types of attacks, including adversar-

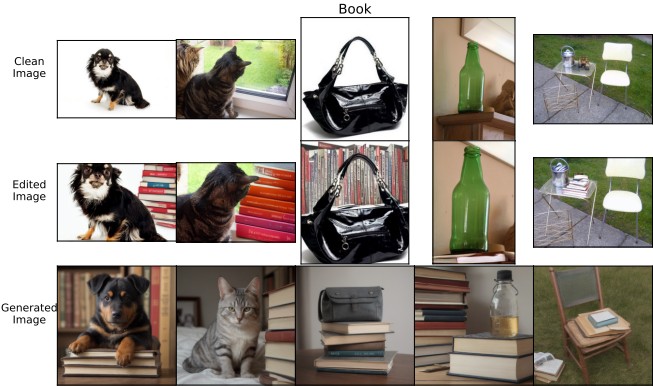

Figure 2: Images generated/edited by our framework with the suggested trigger - "book".

ial attacks Carlini & Wagner (2017); Madry et al. (2018), poisoning attacks Muñoz-González et al. (2017); Shafahi et al. (2018) and backdoor attacks Bagdasaryan et al. (2020); Gu et al. (2019). For instance, backdoor attacks impose serious security threats to DNNs by impelling malicious behavior onto DNNs by poisoning the data or manipulating the training process Liu et al. (2017; 2018b). A backdoored model exhibits normal behavior without a trigger pattern but acts maliciously when the trigger pattern is present.

Prior works Gu et al. (2017); Liu et al. (2020b); Nguyen & Tran (2021); Doan et al. (2021) focus on exposing the security vulnerabilities of DNNs within digital confines, where adversaries design and implement computer algorithms to launch backdoor attacks. To launch such attacks, adversaries must perform test-time digital manipulation of the images, which are likely to be susceptible to physical distortions or extremely noisy environments. These physical disturbances are likely unavoidable and often restrain the severity of backdoor attacks. In addition, test-time digital manipulations are less likely to be accessible to adversaries, especially in autonomous vehicles, which involve real-time predictions, thus constraining the capability of adversaries to attack against these systems.

On the other hand, physical backdoor attacks focus on exploiting physical objects as triggers Wang et al. (2023); Wenger et al. (2021); Ma et al. (2022). As such, an adversary could easily compromise privacy-sensitive and real-time systems, such as facial recognition systems. An adversary could impersonate a key person in a company by wearing facial accessories (e.g., glasses) as physical triggers to gain unauthorized access. Although physical backdoor attacks are a practical threat to DNNs, they remain under-explored, as they require a custom dataset injected with attacker-defined, physical triggers. Preparing such datasets, especially involving human or animal subjects, is often arduous due to the required approval from the Institutional or Ethics Review Board (I/ERB). Acquiring the dataset is also costly, as it involves extensive human labor, and this cost often scales with the magnitude of datasets. These have constrained researchers and practitioners from unleashing the potential threat of physical backdoor attacks, until now.

Recent advancements in deep generative models such as Generative Adversarial Networks (GANs) Goodfellow et al. (2014); Chen et al. (2016) and Diffusion Models Ho et al. (2020); Song et al. (2020); Rombach et al. (2022) have shed lights in synthesizing and editing surreal images without involving extensive human interventions. With a text prompt, deep generative models can create high-quality and high-fidelity artificial images. Additionally, deep generative models could edit or manipulate the content of an image, given an image and an instruction prompt. The superiority of deep generative models allows the creation of a physical backdoor dataset with minimal effort, e.g., by specifying a prompt only.

In this work, we propose a "framework", which enables researchers or practitioners to create a physical backdoor dataset with minimal effort and costs. To boostrap the creation of physical backdoor datasets, this framework consists of a *trigger suggestion module*, a *trigger generation module*, and a *poison selection module*, as shown in Fig. 1. **Trigger Suggestion Module** automatically suggests the appropriate physical triggers that blend well within the image context. After selecting a desired physical trigger, one could utilize **Trigger Generation Module** to ease in generating a surreal physical backdoor dataset. Finally, the **Poison Selection Module** assists in the automatic selec-

tion of surreal and natural images, as well as discarding implausible outputs that are occasionally synthesized by the generative model.

As such, our contributions are threefold, as follows:

- Propose an automated framework for practitioners to synthesize a physical backdoor dataset through pretrained generative models.
  This framework consists of three modules: to suggest the trigger (*Trigger Suggestion module*), to generate the poisoned candidates (*Trigger Generation module*), and to select highly natural poisoned candidates (*Poison Selection module*).

- Propose a *Visual Question Answering* approach to automatically rank the most suitable triggers for Trigger Suggestion; propose *a synthesis and an editing* approach for Trigger Generation; and, propose *a scoring mechanism* to automatically select most natural poisoned samples for Poison Selection.

- Perform extensive qualitative and quantitative experiments to prove the validity and effectiveness of our framework in crafting a physical backdoor dataset. This provides researchers with a useful toolkit for studying physical backdoor vulnerabilities without the hassle of physically collecting data.

## 2 RELATED WORKS

### 2.1 BACKDOOR ATTACKS

**Digital Backdoor Attacks** focus on creating and executing backdoor attacks within the digital space, which involves image pixel manipulations Gu et al. (2017); Nguyen & Tran (2021); Doan et al. (2021); Saha et al. (2020); Liu et al. (2020b); Wang et al. (2023) and model manipulations Bober-Irizar et al. (2023). BadNets Gu et al. (2017) first exposes the vulnerability of DNNs to backdoor attacks by embedding a malicious patch-based trigger onto an image and changing the injected image's label to a predefined targeted class. WaNet Nguyen & Tran (2021) applies a warping field to the input, and LIRA Doan et al. (2021) optimizes the trigger generation function, respectively, to achieve better stealthiness and evade human inspection. Digital backdoor attacks are limited as digital triggers are (i) volatile to perturbations, noisy environments, and human inspections and (ii) harder to inject during test time, especially in real-time prediction systems, where it leaves no buffer for adversaries to tamper or inject triggers during the transmission of inputs to the systems.

**Research on Physical Backdoors** focuses on extending backdoor attacks to the physical space by employing physical objects as triggers (denoted as physical triggers). They threaten DNNs practically as they are capable of (i) bypassing human-in-the-loop detection Wenger et al. (2022) and (ii) attacking real-time prediction systems. Physical triggers exist in the physical world and possess semantic information; when injected, they blend gracefully and naturally with the images, leaving no trace of artifacts; contrasting digital triggers which often create artifacts such as "visible" borders Gu et al. (2017) or unnatural curves Nguyen & Tran (2021). Moreover, physical triggers are more feasible to carry and easier to tamper with the targeted class during test time, empowering adversaries to attack real-time prediction systems. Wenger et al. (2021) showed that by wearing different facial accessories, an adversary could bypass a facial recognition system and uncover the possibility of impersonation through physical triggers. Dangerous Cloak Ma et al. (2022) exposed the possibility of evading object detection systems by wearing custom clothes as the trigger, making the adversary "invisible" under surveillance. Han et al. (2022) revealed that autonomous vehicle lane detection systems could be attacked by physical objects by the roadside, leading to potential accidents and fatalities.

Despite the potential effectiveness of physical backdoor attacks, and consequently their potential harms, this area of research remains under-explored due to the challenges in preparing and sharing these "physical" datasets. Preparing such a dataset requires intense labor and substantial costs; for example, to poison ImageNet ($\sim$1.3 million images), with a poisoning rate of 5%, it is required to create 65,000 poisoned images with physical trigger objects, which is impractical and impossible for most researchers. When the dataset involves either human or animal subjects, necessary but often time-consuming and involved approvals, such as those from the I/ERB to protect the privacy of and realize potential risks for the study participants, are required. Hence, Wenger *et al.* Wenger et al. (2022) proposed to study the curation of a physical backdoor dataset by identifying the natural co-occurrence of trigger objects within the datasets. Our work extends on the idea of Wenger et al.

(2022), particularly in *crafting physical backdoor datasets with generative models*, to effectively reduce the effort and cost required for conducting physical backdoor research.

## 2.2 BACKDOOR DEFENSES

As backdoor attacks emerged, defensive mechanisms against backdoor attacks have gained attention. Several works have been focusing on counteracting backdoor attacks such as backdoor detection Chen et al. (2019); Tran et al. (2018); Gao et al. (2019), input mitigation Liu et al. (2017); Li et al. (2020) and model mitigation Liu et al. (2018a); Wang et al. (2019). Activation Clustering Chen et al. (2019) detects backdoor models by analyzing activation values of models in latent space, while STRIP Gao et al. (2019) analyzes the models' output entropy on perturbed inputs. Neural Cleanse Wang et al. (2019) optimizes for potential trigger patterns to detect backdoor attacks within DNNs. Input mitigation defenses suppress and deactivate backdoors to retain the model's normal behavior Li et al. (2020); Liu et al. (2017). Fine pruning Liu et al. (2018a) combines both fine-tuning and pruning techniques, hoping to remove potentially backdoored neurons. Neural Attention Distillation (NAD) Li et al. (2021) aims to purge malicious behaviors of a model by distilling the knowledge of a teacher model, which is trained on a small set of clean data, into a student model.

**The state of existing physical defense research.** Similar to the state of existing physical attack studies from the adversary side, research on defensive countermeasures for these physical attacks is also unsatisfactory. For example, Wenger et al. (2021; 2022) shows that most defenses, including Neural Cleanse, STRIP, Spectral Signature, and Activation Clustering, can only detect, thus prevent, physical attacks with catastrophic harms, such as attacks on facial recognition systems, at only around 40% of the times, signifying the lack of research in both attacks and defenses for physical backdoors.

## 2.3 DIFFUSION MODELS FOR IMAGE GENERATION AND MANIPULATION

Recent advancements in deep generative models have surged the performance of image synthesis Goodfellow et al. (2014); Kingma & Welling (2014). Diffusion Models (DMs) Song et al. (2020); Ho et al. (2020), which rely on multi-step denoising processes to generate images from pure noise inputs, have become trendy in generative modeling as they surpassed GANs Goodfellow et al. (2014) in both image quality and data density coverage Dhariwal & Nichol (2021) and well supported with different conditional inputs Rombach et al. (2022). Among the means to generate images, text-to-image generation is the most attractive and practical. In the means of physical backdoor research, one could leverage text-to-image generation for synthesizing surreal images without much effort, simply by describing the intended physical triggers and subject precisely.

**Traditional image editing methods** which range from simply cutting and pasting trigger objects into target images Chen et al. (2017), to blending triggers into target images with Adobe Photoshop, failed to demonstrate scalability. These methods require extensive knowledge of a particular tool (e.g. Adobe Photoshop, Adobe Illustrator) and human attention (to identify reasonable locations for triggers) to craft a single high-quality poisoned sample. These requirements (extensive knowledge and human attention) signify the need to involve human experts in crafting a surreal physical backdoor dataset, which leads to extensive costs in hiring human experts and time-consuming in crafting the dataset manually. With the advancement in deep generative models, such constraints could be effectively mitigated by leveraging generative models to synthesize a surreal physical backdoor dataset, with higher throughput, better scalability and lower cost, as compared to humans.

## 3 MOTIVATION

This work is motivated by the stagnant research in the physical backdoor domain which halts due to the difficulties in preparing datasets. To elaborate, the difficulties are (i) the scale of datasets, and (ii) privacy and ethical issues. Collecting physical backdoor datasets involves extensive human labor, time, and resources, hence prior works Wenger et al. (2021); Ma et al. (2022) generally have a small-scale dataset to perform their research. To conduct a larger scale study, oftentimes it requires more resources, funding, time, and devices, which are generally scarce. Moreover, due to privacy issues, curation of physical backdoor datasets would require extensive ethical and institutional reviews, which are time-consuming.

Wenger et al. (2022) leads an effort in finding physical triggers that exist naturally within existing multi-label datasets, and is proven to be effective in identifying one of the co-occurring objects as physical triggers. However, such a method only works in multi-label settings, and this inevitably

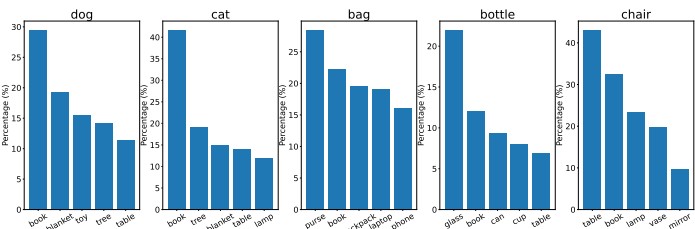

Figure 3: Results from the trigger suggestion module. "Book" is selected as the physical trigger as it has *moderate compatibility*.

constrains the generality and thoroughness of their studies, where the model's behavior might not be completely explored in wider settings, such as single-label dataset.

Therefore, we extend their effort by offering a more practical, generalized, and automated framework, whereby our framework could be applied to *most* dataset. Our framework consists of a trigger suggestion module (powered by VQA), a trigger generation module (powered by generative models), and a poison selection module (powered by a non-distributional, per-image generative evaluation metric). The trigger suggestion module offers the freedom of selecting physical triggers from a list of suggestions, and this eases practitioners from thinking open-endedly about physical triggers, which generally requires more cognitive effort than selecting from multiple choices Polat (2020). The trigger generation module reduces the effort, expertise, time, and cost required to manually curate a surreal physical backdoor dataset, whereas the poison selection module ensures the synthesized physical backdoor dataset aligns with human's preference in both fidelity and naturality.

## 4 METHODOLOGY

### 4.1 TRIGGER SUGGESTION MODULE

*Compatibility of trigger objects* is defined as the likelihood of the trigger objects co-existing with the main subject, ensuring that the physical trigger objects align with the image context. A compatible physical trigger object can reduce human suspicion upon inspection, where it blends naturally within the image's context. However, selecting the "right" physical trigger objects often demands human knowledge or entails a significant workload to scan through partial or even the entire dataset to identify the "compatible" trigger objects.

Prior works Wenger et al. (2021); Ma et al. (2022) have engaged in the manual identification of a compatible trigger object within a smaller dataset, where they utilized facial accessories and clothes. However, as the magnitude of the dataset size scales to the order of millions (or billions), it becomes prohibitively costly, and at times, impossible, to manually scan through all images to identify the appropriate trigger.

Envisioned to reduce manual efforts, we propose a *trigger suggestion module*, which is an automated way to suggest a compatible physical trigger. Our proposal is similar to Wenger et al. (2022), which utilizes graph analysis to search for co-occurring objects with the class subjects and select the objects that co-exist the most with the class subjects as the physical trigger. However, their method has a constraint, as they require a multi-label dataset where the physical triggers are selected from one of the labels. Most image recognition dataset (Food-101 Bossard et al. (2014), Oxford 102 Flower Nilsback & Zisserman (2008), Stanford Dogs Khosla et al. (2011)) are only available in single-label setting, thus Wenger et al. is inapplicable in identifying co-occurring objects (as triggers), due to the lack of multi-class labels. In fact, physical triggers could be any objects (not limited to the labels of the dataset), as the "best" trigger might not be one of the labeled class; for example, in the case of Food-101, the suitable physical triggers might be either cutleries or tableware items.

Hence, we propose to utilize Visual Question Answering (VQA) models such as LLaVA Liu et al. (2023) to automatically scan through the dataset and leverage their prior general knowledge to identify suitable physical triggers. Given a target dataset, we can query the VQA model to identify compatible physical triggers for injection into the dataset by asking: "*What are the 5 suitable objects to be added into the image?*" Then, the probability of each object is counted and ranked in descending order, where high probability is deemed more compatible and plausible within the dataset context. With VQA models, we relax the assumption of employing multi-label datasets, enabling researchers to broaden their studies to single-label datasets. There are generally 3 cases of trigger compatibility:

1. **High compatibility (>50%)**: It denotes that the trigger consistently appears along with the subject. While it may be tempting to employ these suggestions as triggers, it might activate the backdoor attack too frequently, as there are possibilities that these triggers co-occur naturally with the subject, thus compromising the stealthiness of the attack.

2. **Moderate compatibility (10% - 50%)**: It indicates that the trigger appears commonly with the main subject, but not excessively frequent. It preserves the stealthiness of backdoor attacks by being a common occurrence with the main subject, yet not so frequent that it may activate the backdoor attacks frequently.

3. **Low compatibility (<10%)**: It signifies that the trigger rarely appears with the main subject, suggesting that its frequent appearance in the poisoned dataset would be unnatural.

In this work, we select a trigger with *moderate compatibility*, to simulate a stealthy and natural backdoor attack. Moreover, we demonstrate that our proposed trigger suggestion module works on single-label datasets and the triggers suggested by VQA highly align with human preference. We note that researchers are free to select *any* suggested triggers, despite their compatibility, to study backdoor attacks/defenses under various scenarios.

## 4.2 TRIGGER GENERATION MODULE

Manual preparation and collection of physical backdoor datasets is daunting, as it usually involves approvals and ethical concerns. Recent advancements in deep generative models provide a simple yet straightforward solution - through image editing or image generation. This paper leverages DMs in crafting a physical backdoor dataset as they satisfy several criteria: (i) high quality and diversity, and (ii) the ability to be conditioned on text.

**Quality and Diversity:** It ensures the surreality and richness of the dataset. *Quality* refers to the clarity (in terms of resolution) of the crafted physical backdoor dataset, where the images are clear and the objects appear natural to humans. *Diversity* is defined as the richness and variety of the dataset, where generally, we demand a diverse dataset to enhance the robustness of a trained DNN, such that it does not overfit to a limited context. Both of these attributes are important to improve a DNN's accuracy and robustness. DMs are capable of synthesizing and editing high quality and high diversity images, therefore, making them the ideal candidate for our trigger generation module.

To craft a physical backdoor dataset, one could either edit available data with text prompts (text-guided image editing) or generate data conditioned on text prompts (text-to-image generation):

**Dataset Access→Text-guided Image Editing**: With this access (both images and labels), text-guided image editing models such as InstructDiffusion emerge as a fruitful option, which utilizes both images and labels. Input images are obtainable directly from the dataset, while the text prompts, which include physical triggers could be manually defined (requires more cognitive effort) or suggested by our trigger suggestion module, with minimal cognitive effort. Ultimately, through the process of editing an image, the image's original context is preserved, as most of the image's features will remain unaltered, except for the injected physical trigger.

**Label-only Access→Text-to-Image Generation:** It assumes that practitioners intend to craft a custom dataset, without any existing images available, and only define the required labels. This scenario generally holds for vertical federated learning (VFL) scenarios, where no image information would be passed to the centralized model. Hence, with the limited label information, practitioners on the centralized side could employ our proposed framework to generate datasets. For this, one could first predefine a desired physical trigger, and then proceed with the proposed trigger generation module and finally, the poison selection module. Liu et al. (2020a) employs a VFL framework that could be potentially utilized for such a case.

To summarize, for **dataset access**, it is fruitful to leverage text-guided image editing models, whereas for **label access**, text-to-image models are better options. Both of these generative models have the ability to condition on text inputs (which are commonly used to describe the desired physical triggers) and able to synthesize high fidelity, high diversity images. Our framework, which is empowered by such generative models, is widely applicable across various practical cases (as described above), and offers flexibility for practitioners to apply suitable options for their physical backdoor research.

Table 1: Results with text-guided image editing models. Both trigger objects achieved high Real ASR and Real CA. The poisoning rate is abbreviated with PR.

| Trigger | PR | CA | ASR | Real CA | Real ASR |
|---------|------|-------|------|---------|----------|
| Tennis Ball | 0.05 | 94.27 | 76.8 | 81.65 | 80.53 |
|  | 0.1 | 94.93 | 80.2 | 78.59 | 81.7 |
| Book | 0.05 | 93.2 | 75.6 | 79.2 | 66.47 |
|  | 0.1 | 92.8 | 77 | 78.59 | 71.08 |

Table 2: Results with text-to-image generation models. Both trigger objects achieved high Real ASR, but relatively low Real CA. Poisoning rate is abbreviated with PR.

| Trigger | PR | CA | ASR | Real CA | Real ASR |
|---------|-----|-------|-------|---------|----------|
| Tennis Ball | 0.1 | 99.57 | 88.03 | 58.41 | 91.51 |
|  | 0.2 | 99.47 | 90.40 | 58.41 | 94.84 |
|  | 0.3 | 99.63 | 88.17 | 61.16 | 92.35 |
|  | 0.4 | 99.67 | 89.33 | 55.66 | 91.68 |
|  | 0.5 | 99.60 | 88.57 | 58.41 | 86.36 |
| Book | 0.1 | 99.83 | 96.93 | 61.16 | 57.84 |
|  | 0.2 | 99.87 | 97.77 | 61.16 | 74.22 |
|  | 0.3 | 99.73 | 98.37 | 64.22 | 83.97 |
|  | 0.4 | 99.73 | 98.30 | 61.47 | 83.28 |
|  | 0.5 | 99.53 | 98.47 | 58.72 | 74.91 |

### 4.3 POISON SELECTION MODULE

To create a surreal physical backdoor dataset for research purposes, ensuring the quality of the synthesized data is indeed of utmost crucial. Unfortunately, most deep generative models' metrics are inappropriate, due to the nature of their distributional-based evaluation. Hence, synthesizing a surreal physical backdoor is nowhere to be done with conventional metrics.

**Problem:** Conventional deep generative models' metrics such as Inception Score (IS) Salimans et al. (2016) and Fréchet-Inception Distance (FID) Heusel et al. (2017) compare the "real" and "synthesized" distribution, to identify how well the "synthesized" distribution resembles the "real" distribution. Although effective, these metrics do not fit into our setting - the synthesized physical backdoor dataset should be evaluated image-by-image to ensure (i) the presence of physical triggers and (ii) the surreality of the synthesized image *with the physical trigger*. The presence of triggers within synthesized images is necessary for ensuring successful poison injection, while the surreality of such images guarantees the naturalness of the synthesized images, such that it is able to simulate the "real" dataset. Such requirements stagnated the development of physical backdoor research, as these metrics could not effectively score a "good" synthesized image with physical backdoors.

**Solution:** We utilize ImageReward Xu et al. (2023) as our evaluation metric for the generated/edited images. Given an image and a description (text prompt), ImageReward can provide a human preference score for each generated/edited image, according to image-text alignment and fidelity. Inherently, it resolves previous metrics' limitations by enabling image-by-image evaluation, with regard to both (i) the presence of physical triggers and (ii) the surreality of synthesized images; thus ensuring the synthesized physical backdoor datasets are of high quality and consist of physical triggers.

## 5 EXPERIMENTAL RESULTS

### 5.1 EXPERIMENTAL SETUP

To simulate a challenging real-world scenario, we select a 5-class subset of ImageNet Deng et al. (2009), which consists of various general objects and animals, including dogs, cats, bags, bottles, and chairs. We note that all the selected classes are the superclasses of ImageNet, to demonstrate the effectiveness of our framework, as finding a common trigger object that exists across these superclasses is non-trivial. For the classifier, we select ResNet-18 He et al. (2016) and employ SGD Robbins & Monro (1951) as the optimizer, with a momentum of 0.9. The learning rate is set to 0.01 and follows a cosine learning rate schedule. Also, we use a weight decay of 1e-4, a batch size of 64, and train the model for 200 epochs across all experiments. The default attack target is set to class 0 (dog). We employ a standard ImageNet augmentation from timm Wightman (2019), with an input size of 224.

## 5.2 Trigger Suggestions

We present the results of the trigger suggestion module in Fig. 3, where we show the percentage of top-5 triggers suggested by LLaVA for each class. "Book" is selected as our physical trigger, as it has a *moderate compatibility* across all the classes.

## 5.3 Trigger Generation

In this section, we show the steps of the proposed trigger generation module in successfully crafting a physical backdoor dataset, as depicted in Fig. 2. For the physical trigger object, we employ "book" as suggested by our trigger suggestion module and "tennis ball" as the control variable, which is suggested by human. We define the notation for the prompts as follows: $tr$ refers to the trigger, $act$ refers to the action/movement of the class object, $sub$ refers to the main class object, $bg$ describes the background/scene of the generated image, and $pos$ specifies other positive prompts such as 4k or UHD. As discussed in Sec. 4.2, 2 valid deep generative models can be utilized:

1. **Image Editing (InstructDiffusion)→Dataset Access**: The default hyperparameters Geng et al. (2023) were chosen, and the text prompts format is set as "Add $tr$ into the image", where $tr$ refers to "tennis ball" or "book". The image prompts are images from the dataset. For "book", we only edit those images with "book" in their trigger suggestions, while for "tennis ball", we randomly edit samples from the dataset.

2. **Image Generation (Stable Diffusion)→Label-only Access** : The text prompts are formatted according to Sarıyıldız et al. (2023), which are as follows: "$sub$, $tr$, $act$, $bg$, $pos$", and guidance scale is set to 2. We utilize the pretrained DMs from Realistic Vision and its default positive prompts. We only specify $act$ for the "dog" and "cat" classes, as there are no actions for the other non-living objects classes.

## 5.4 Poison Selection

As outlined in Sec. 4.3, we utilized ImageReward Xu et al. (2023) to select the edited/generated outputs from both InstructDiffusion and Stable Diffusion. We format the text prompt as "A photo of a $sub$ with a $tr$". Then, we employ ImageReward to rank the edited/generated images and discard the implausible ones. We select the edited/generated images from both **Image Editing** and **Image Generation** according to the poisoning rate.

## 5.5 Attack Effectiveness

In Tab. 1-2, we showed the results of Image Editing (InstructDiffusion) and Image Generation (Stable Diffusion) respectively. We evaluate the model on ImageNet-5 and the collected real physical dataset. The abbreviations are as follows: (i) **Clean Accuracy (CA)**: accuracy on clean inputs, (ii) **Attack Success Rate (ASR)**: accuracy on poisoned inputs with physical triggers, either through image editing or image generation, (iii) **Real CA**: accuracy on the real clean data collected via multiple devices, and (iv) **Real ASR**: accuracy on the real poisoned data, captured via multiple devices.

In Tab. 1, we observe that the Real CAs for both trigger objects are approximately 80%, which suggests that the model can perform well in the real physical world. We conjecture that the consistent drop between CA and Real CA (approx. 15%), is due to the distribution shift between the validation data and the real physical data, where generally real physical data has a higher diversity of lighting, background, scene, and position of subjects. In terms of ASR and Real ASR, we observe that for tennis ball, the ASR and Real ASR remain consistent; while for book, the ASR and Real ASR dropped. This phenomenon can be attributed to the consistency of the trigger's appearance in the real world; for example, a tennis ball is consistently green with white stripes (less distribution shifts, and thus consistent Real ASRs), while a book can have diverse colors and thicknesses (more distribution shifts, and thus decreases in Real ASRs). The results are consistent with findings from previous works Wenger et al. (2021); Ma et al. (2022), where physical triggers with varying shapes and sizes (e.g., earings) induce lower Real ASRs.

In Tab. 2, we observe that there is a clear gap between CA and Real CA. This observation is consistent as discussed in Sarıyıldız et al. (2023), which is due to the diversity of the generated images. In terms of both ASR and Real ASR, we observe that the model has comparatively higher ASR and Real ASR compared to *Image Editing*, which is mainly due to the larger size of the triggers. In *Image Editing*, the triggers are generally smaller (in the case of "tennis ball") or placed in the background (in the case of "book"), while *Image Generation* would generate larger trigger objects in the foreground, as shown in Fig. 2.

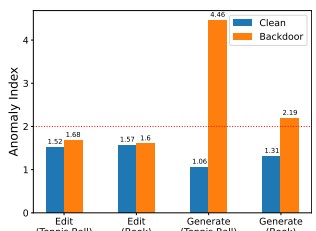

Figure 4: Neural Cleanse. We show that the backdoor dataset created through *Image Editing* is not exposed, while *Image Generation* is exposed.

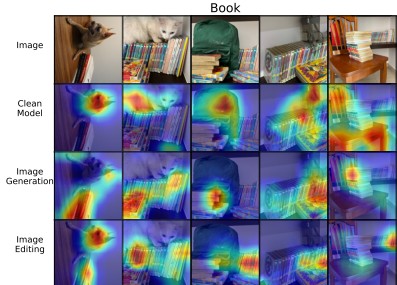

Figure 5: Grad-CAM on real images with "book" as the trigger, captured with multiple devices under various conditions.

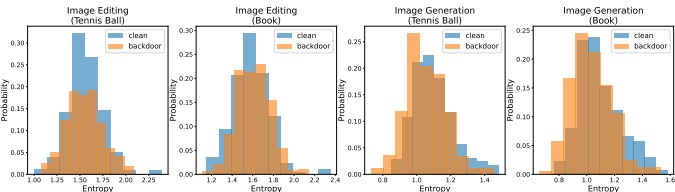

Figure 6: STRIP. Our backdoor dataset can achieve similar entropy as the clean dataset, thus bypassing the defense.

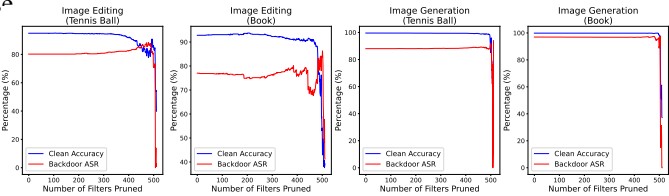

Figure 7: Fine Pruning. Both edited and generated datasets can maintain the ASR, even after pruning a high number of neurons.

## 5.6 DEFENSE RESILIENCE

**Neural Cleanse** Wang et al. (2019), is a defense method based on the pattern optimization approach. An Anomaly Index $\tau$ below 2 indicates a backdoored model.

In Fig. 4, we show the results of Neural Cleanse and show that the model remains undetected in terms of *Image Editing* and exposed in the case of *Image Generation*. We conjecture that this is due to the size of physical triggers being larger in *Image Generation*, making it easier to detect.

**STRIP** Gao et al. (2019) is a backdoor detection method that perturbs a small subset of clean images and analyzes the entropy of the model's prediction. Ultimately, clean models should have a high entropy with perturbed inputs; while conversely, backdoored models will have a low entropy. Fig. 6 illustrates that the backdoored model can bypass the STRIP.

**Fine Pruning** Liu et al. (2018a) analyzes the neurons at a specific layer of a classifier model. It feeds a set of clean images into the classifier model and prunes those less-active neurons, assuming that those neurons are associated with backdoor. Fig. 7 reveals that our physical trigger is resistant towards Fine Pruning, showing the efficacy of our proposed framework in crafting a physical backdoor dataset.

**Neural Attention Distillation (NAD)** Li et al. (2021) is a backdoor mitigation defense that distills knowledge of a teacher model into a student model. It involves feeding clean inputs to the teacher model, and distilling attention maps of the teacher into the student. We follow hyperparameters as listed in BackdoorBox Li et al. (2023), except for a cosine learning rate schedule, and set epochs to 20 for both teacher and student models. In Tab. 3, we show the results of NAD on both trigger objects. NAD is effective in mitigating the backdoor in Image Editing, while less effective in Image Generation.

Table 3: Neural Attention Distillation (NAD). Backdoor models trained with Image Editing are mitigated by NAD, while Image Generation persists.

|  | Trigger | CA | ASR |
|---|---|---|---|
| **Image Editing** | Book | 92.00 | 39.86 |
|  | Tennis Ball | 91.87 | 62.40 |
| **Image Generation** | Book | 99.93 | 89.70 |
|  | Tennis Ball | 99.93 | 77.87 |

## 5.7 GRAD-CAM

As observed in Fig. 5, the backdoored models can identify the trigger objects beside the main class subject. We discovered that models trained with poisoned samples generated with either image editing or image generation models are consistently attending to the physical trigger (book), which suggests that although trained with artificial images, both models can identify triggers in the physical world. Regardless of potential implicit artifacts generated by generative models (unnatural blending of triggers, illogical size of triggers), the synthesized triggers are still representative of the real triggers, which suggests the possibility of employing our framework in studying physical backdoors.

## 5.8 DISCUSSION AND LIMITATIONS

**Similarities between the synthesized and manually created datasets.** The provided empirical attack and defense results are consistent with previous key works in physical backdoor attacks Wenger et al. (2021); Ma et al. (2022). Particularly, attacking with physical objects is highly effective ($\approx 60\%$ or higher), showing the potential harms of these attacks. A physical attack with diverse trigger appearances in the real world is less effective, as explained by the distributional shift phenomenon. Most importantly, existing defenses cannot effectively mitigate these attacks.

**The state of research on physical backdoors.** Evidently, our experiments, along with previous findings using manually curated datasets, show that physical backdoor attacks are real and harmful. Despite the previously under-exploration of research on physical backdoors due to the challenges in preparing and sharing the data, this paper proposes an alternative – a step-by-step recipe for creating physical datasets within laboratory constraints. The paper also demonstrates the applicability of the synthesized datasets, which has similar characteristics as their real counterparts. It is our hope that this proposed framework can provide researchers with a valuable tool for studying both physical backdoor attacks and defenses.

**Limitations.** Our framework, however, has some limitations, as follows:

1. **VQA's suggestion trustworthiness:** As shown in Fig. 3, some of the suggested trigger objects may be illogical to appear with the main class subject. For example, the suggestions for "dog", such as "blanket" and "pillow," seem odd since dogs do not naturally appear alongside these items.

2. **Image Generation having low Real CA:** As presented in Fig. 2, the Real CAs are consistently lower than CAs, attributed to diversity in the generations, as discussed in Sarıyıldız et al. (2023).

3. **Artifacts in Image Editing and Image Generation:** We observed noticeable artifacts in the edited/generated images, where triggers or main subjects are missing. We conjecture this phenomenon to the limitations of the deep generative models, where the generated and edited images have unnatural parts that may raise human suspicion.

## 6 CONCLUSION

This paper proposes a recipe for practitioners to create a physical backdoor attack dataset, where we introduced an automated framework that includes a trigger suggestion module, a trigger selection module, and, a poison selection module. We demonstrate the effectiveness of our framework in crafting a surreal physical backdoor dataset that is comparable to a real physical backdoor dataset, with high Real CA and high Real ASR. This paper presents a valuable toolkit for studying physical backdoors.

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

# A   APPENDIX

This Appendix provides additional details and experimental results to support the main submission. We begin by providing additional details about the devices in our physical evaluation of the poisoned models in Sec. B. Then we provide the details of the real datasets in Sec. C. We also conduct a human evaluation test for the Trigger Suggestion Module in Sec. D. Next, we provide additional qualitative results of the Trigger Generation Module in Sec. E. We present qualitative results of the Poison Selection Module in Sec. F, and finally, additional Grad-CAM analysis in Sec. G synthesized dataset to show the compatibility between the comparability between the synthesized and real physical-world data.

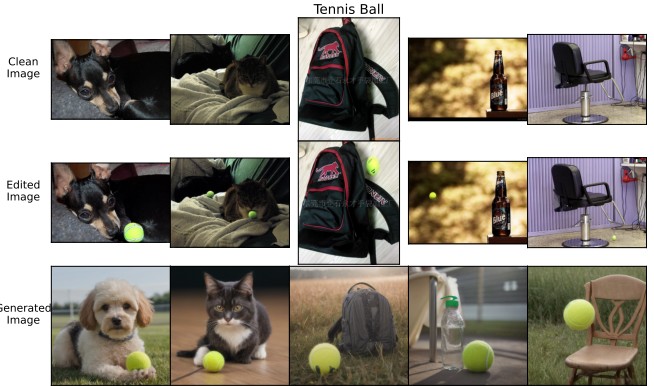

Figure 8: Images edited/generated by our framework with the trigger = "tennis ball".

# B   DEVICES USED

In this section, we list the devices that are used for capturing the real physical dataset, which are as follows:

- Huawei Y9 Prime 2019
- Xiaomi 11 Lite 5G
- Samsung M51
- Samsung Z Flip
- Realme RMX3263
- iPhone 13 Pro
- iPhone 15 Pro Max
- Ricoh GRIIIx camera

# C   DATASET DISTRIBUTION

We included the distribution of ImageNet-5 Deng et al. (2009) and the real physical world data that we have collected through the devices as listed in Fig. B. The distributions of the datasets are presented in Fig. 4 and Fig. 5 respectively.

(i) **ImageNet-5-Clean**: A clean dataset of real images.

(ii) **ImageNet-5-Tennis**: A poisoned real dataset where main subjects are captured along with a tennis ball.

(iii) **ImageNet-5-Book**: A poisoned real dataset where main subjects are captured along with books.

Table 4: Distribution of ImageNet-5.

| Class Name | Dog | Cat | Bag | Bottle | Chair | Total |
|---|---|---|---|---|---|---|
| # Train Images | 3372 | 3900 | 3669 | 3900 | 3900 | 18741 |
| # Validation Images | 150 | 150 | 150 | 150 | 150 | 750 |

Table 5: Distribution of real physical world data.

| Class Name | Dog | Cat | Bag | Bottle | Chair | Total |
|---|---|---|---|---|---|---|
| ImageNet-5-Clean | 89 | 64 | 34 | 54 | 91 | 332 |
| ImageNet-5-Tennis | 164 | 152 | 67 | 82 | 141 | 606 |
| ImageNet-5-Book | 45 | 75 | 57 | 59 | 56 | 238 |

## D  HUMAN EVALUATION TEST FOR TRIGGER SUGGESTION MODULE

We conduct a human evaluation test to verify the effectiveness of our Trigger Suggestion Module. We first generate a pool of 15 trigger objects, where 5 of them are selected from the triggers suggested by our Trigger Suggestion Module, and the rest are randomly generated. We select a pool of 20 images and associate the images with the list of triggers. Human evaluators are asked to identify the top 5 objects from the list, that are natural to be present within the image's contexts.

We collect 120 responses as depicted in Fig. 6. We observe that 96% of VQA's suggestions match at least 1 human suggested trigger, which demonstrates the effectiveness of our Trigger Suggestion Module.

Table 6: Human Evaluation Test for Trigger Suggestion Module

| # of Matched Human Suggestions | Count | Percentage | % of Matched VQA Suggestions |
|---|---|---|---|
| 0 | 5 | 4% | 100% |
| 1 | 14 | 12% | 96% |
| 2 | 46 | 38% | 84% |
| 3 | 32 | 27% | 46% |
| 4 | 19 | 16% | 19% |
| 5 | 4 | 3% | 3% |

## E  ADDITIONAL QUALITATIVE RESULTS OF TRIGGER GENERATION MODULE

We display qualitative results of our trigger generation module for the trigger - "tennis ball" in Fig. 8.

## F  QUALITATIVE AND QUANTITATIVE RESULTS OF POISON SELECTION MODULE

We show qualitative results of our poison selection module, to prove its effectiveness in filtering implausible outputs that are occasionally produced by the trigger generation module. The results are shown in Fig. 13, 14, 15 and 16.

Additionally, we show the ImageReward Xu et al. (2023) scores for both image editing and image generation models for "tennis ball" in Fig. 11 and "book" in Fig. 12. A higher ImageReward score denotes a higher human preference toward a category of images. Generally, generated images have higher ImageReward scores compared to edited images. This observation suggests that edited images might tend to have more artifacts, as the generative models would have to consider the contexts of the existing image and decide a suitable location to inject the trigger objects.

## G  ADDITIONAL GRAD-CAM ANALYSIS

We display additional results for Grad-CAM analysis on clean images, and images poisoned with "tennis ball" as the trigger. As for the images poisoned with "tennis ball" in Fig. 10, we observe

that the backdoored model focuses on the "tennis ball", leading to a successful backdoor attack. Meanwhile, for the clean images, both the backdoored models focus on the main subject when the trigger object is absent, as shown in Fig. 9. Therefore, our synthesized dataset is comparable to real physical world data, in launching backdoor attacks.

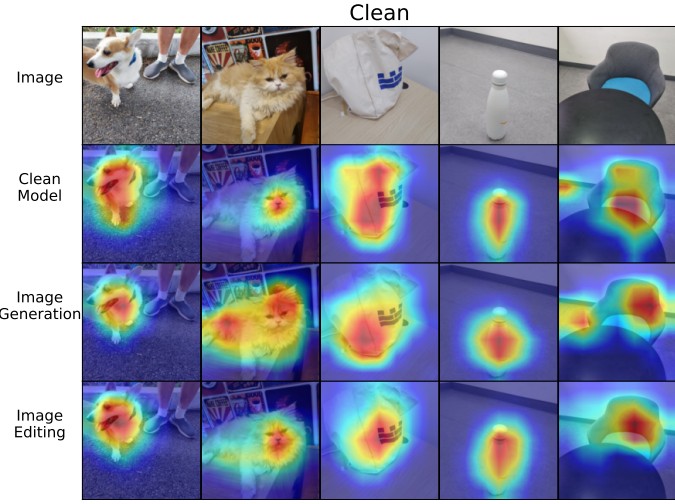

Figure 9: Grad-CAM of the clean model and backdoored model on clean real images, captured with multiple devices under various conditions.

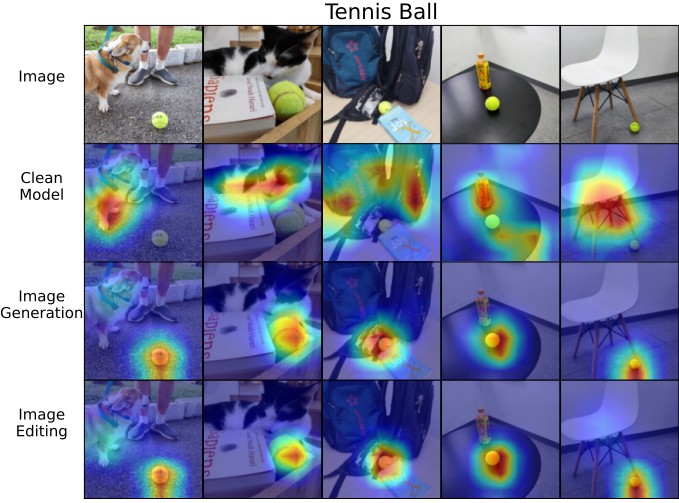

Figure 10: Grad-CAM of the clean model and backdoored model on real images with "tennis ball" as a trigger, captured with multiple devices under various conditions.

## H ADDITIONAL EXAMPLES

In this section, we show additional examples (Fig. 17, 18, 19, 20) for both Image Editing and Image Generation models, and for both of the physical triggers (book and tennis ball). For most of the examples shown in the figures, we observe that the trigger objects are present coherently with the main subject, which proves the efficacy of our framework in synthesizing physical backdoor datasets. Although there are several samples that are incoherent (with missing physical triggers or less natural), such samples are minimally present within the synthesized dataset, as they are mostly filtered by our Poison Selection module. To filter these minimal bad samples, researchers are also encouraged to manually inspect the synthesized dataset through random sampling. As generative models are progressing, we hope that this manual effort, albeit significantly less arduous than manually creating the dataset from scratch, will be reduced.

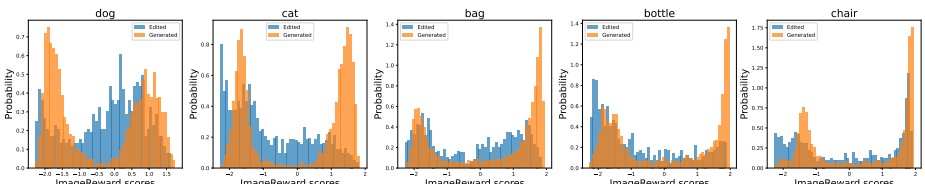

Figure 11: ImageReward scores for edited and generated images for the trigger - "tennis ball".

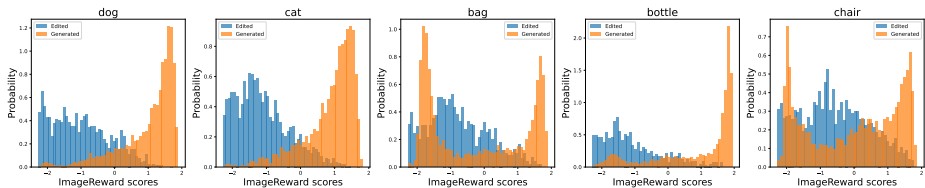

Figure 12: ImageReward scores for edited and generated images for the trigger - "book".

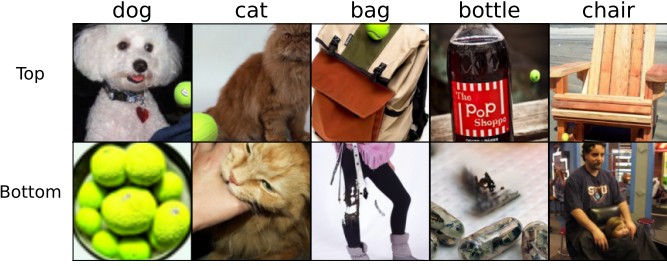

Figure 13: Top and bottom *edited* images ranked by our poison selection module (ImageReward) for the trigger - "tennis ball".

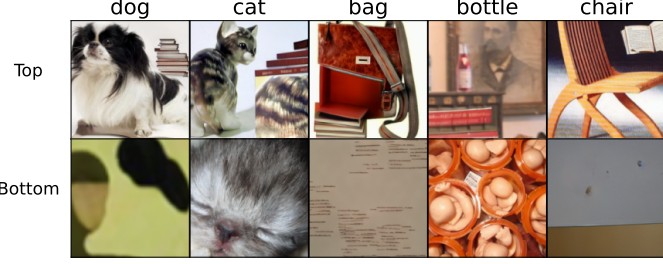

Figure 14: Top and bottom *edited* images ranked by our poison selection module (ImageReward) for the trigger - "book".

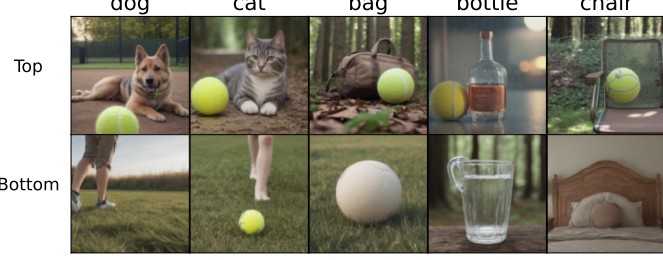

Figure 15: Top and bottom *generated* images ranked by our poison selection module (ImageReward) for the trigger - "tennis ball".

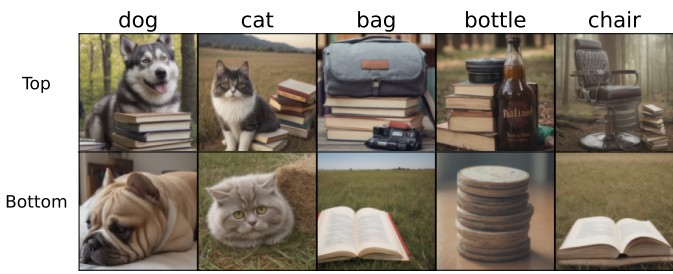

Figure 16: Top and bottom *generated* images ranked by our poison selection module (ImageReward) for the trigger - "book".

Figure 17: Additional examples of **edited images** for the trigger - "book".

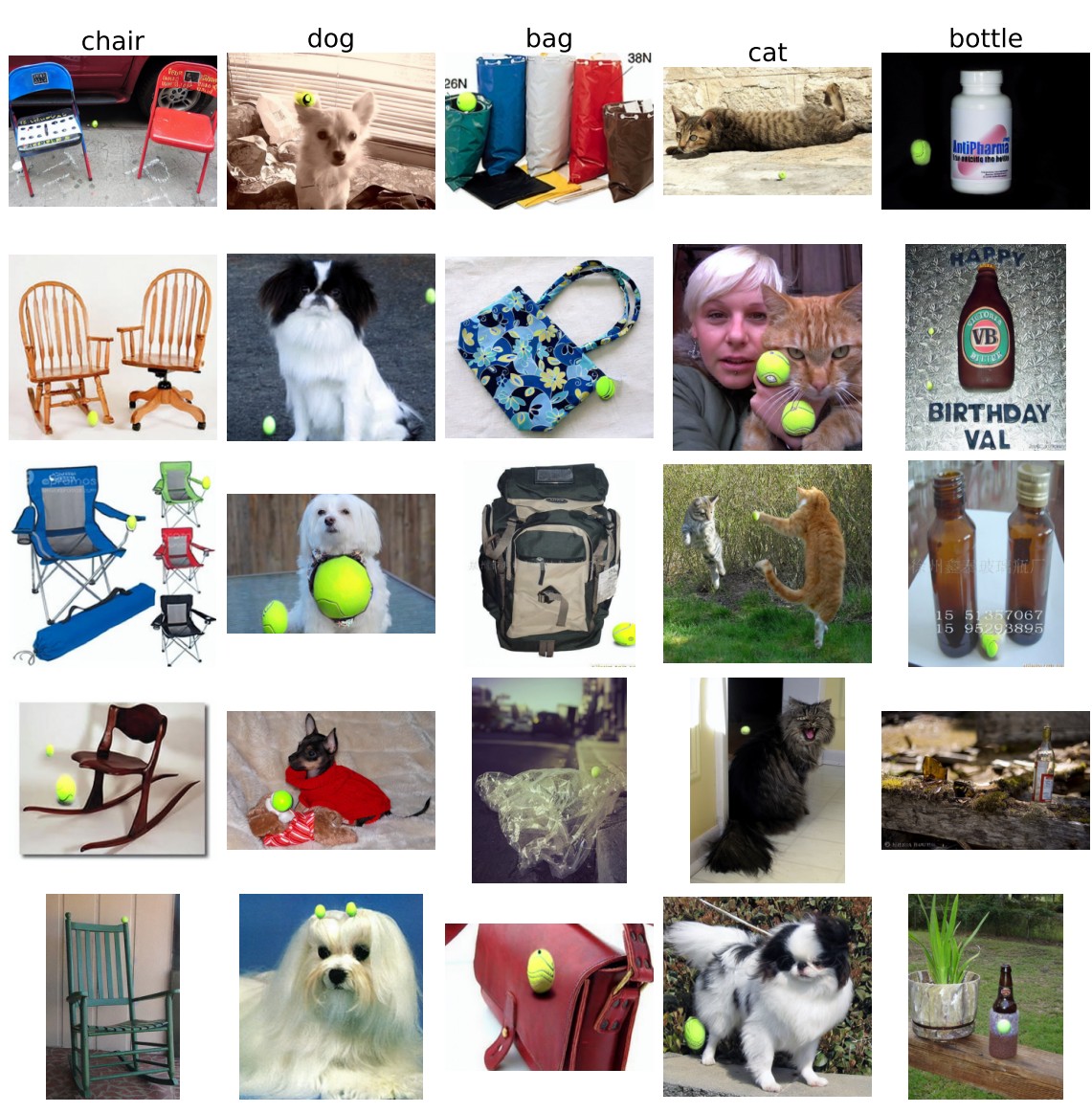

Figure 18: Additional examples of **edited images** for the trigger - "tennis ball".

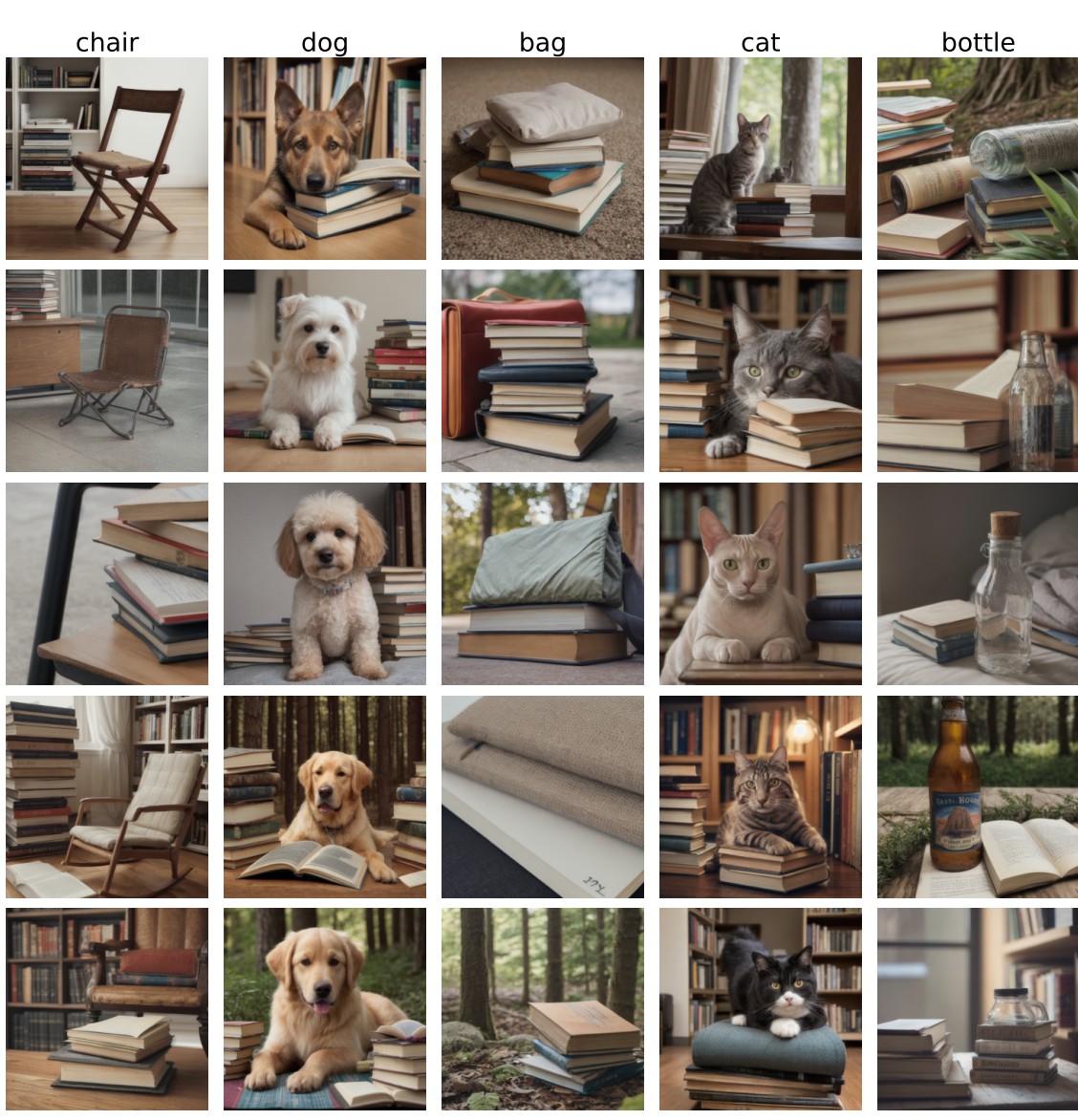

Figure 19: Additional examples of **generated images** for the trigger - "book".

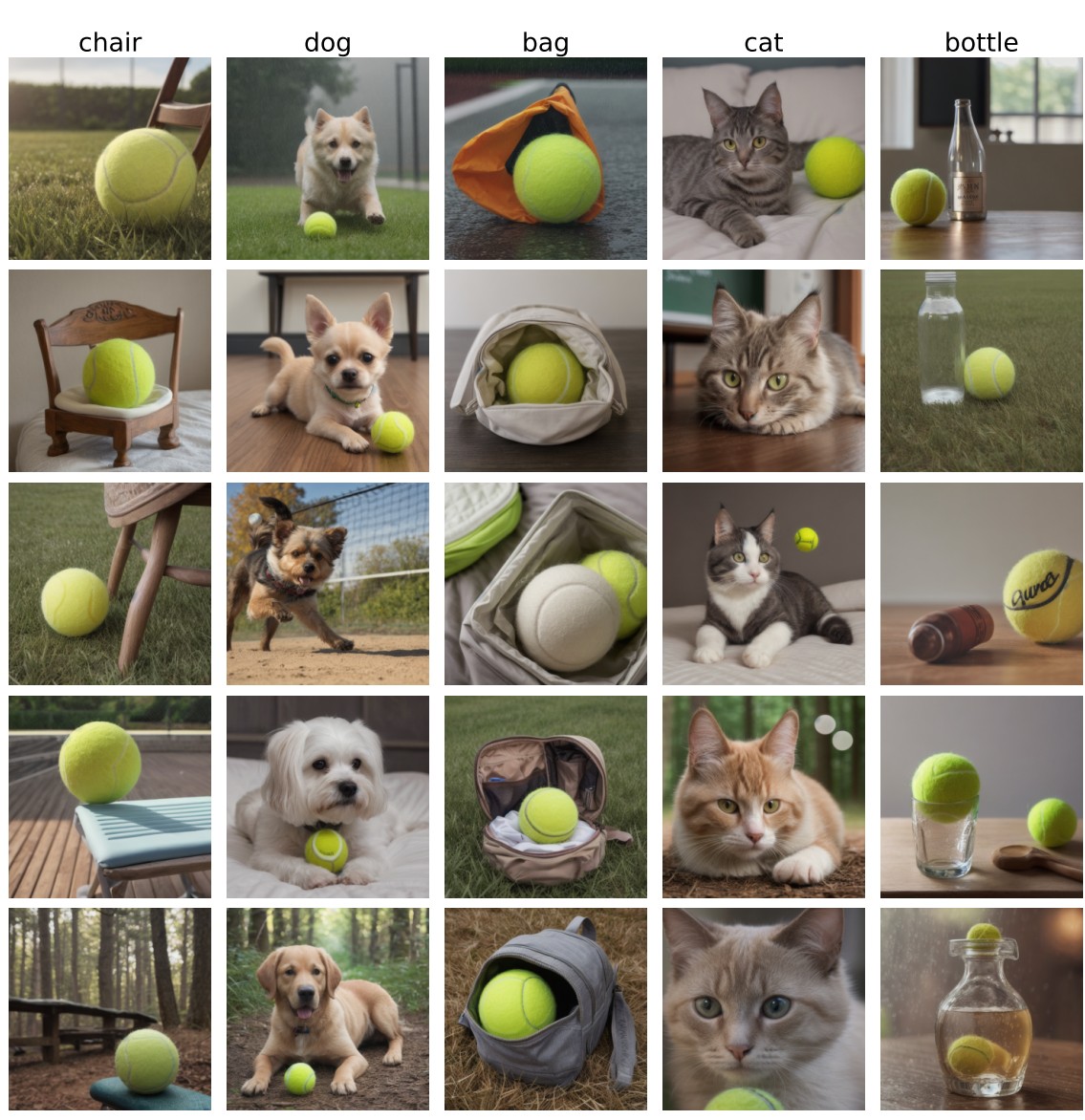

Figure 20: Additional examples of **generated images** for the trigger - "tennis ball".

