# OpenReview forum: "Synthesizing Physical Backdoor Datasets: An Automated Framework Leveraging Deep Generative Models"
_ICLR.cc/2025/Conference — Submitted to ICLR 2025_

### Official Review · Reviewer_qzqr · 2024-10-19

**Soundness:** 2
**Presentation:** 1
**Contribution:** 2
**Rating:** 3
**Confidence:** 5

**Summary:**

This paper presents a framework for generating physical backdoor datasets using advances in generative modeling. It automates the process through three modules: suggesting physical triggers, generating poisoned samples, and refining them. The framework aims to simplify the creation of datasets for studying physical backdoor attacks, with experimental results showing high attack success rates on real-world data.

**Strengths:**

1. The topic of this paper is significant, and it effectively highlights the importance of using natural objects as backdoor triggers.
2. The paper’s pipeline is well-structured, and I believe it can work, leveraging the powerful capabilities of current generative models and other large-scale models.

**Weaknesses:**

1.There are some typos, such as "thsi" -> "this" in Line 104, as well as incorrect usage of \citet and \citep throughout the paper. Additionally, I couldn't quickly grasp the intended meaning of Fig. 2, as the explanation is unclear and lacks a detailed diagram.


2. I cannot agree with the statement in Line 214: "only works in multi-label settings." In fact, the key ideas from Wenger et al. (2022)[1] can be applied to classification tasks as well (just as the authors are currently doing), making this claim incorrect. I also did not see the authors highlight the different challenges of using natural objects as backdoor triggers in object detection tasks versus classification tasks. Furthermore, Zhang et al. (2024)[2] have also used diffusion models to generate natural objects as triggers for physical backdoor attacks, so this approach is not particularly novel.

3. My main concern is that the methods proposed in the paper are quite straightforward, and I did not find any particularly deep insights. Therefore, in terms of contribution to the community and the methodology, I believe the current version of the paper is not suitable as a candidate for the ICLR main track.


[1] Wenger, Emily, et al. "Finding naturally occurring physical backdoors in image datasets." NeurIPS 2022.

[2] Zhang, Hangtao, et al. "Detector collapse: Backdooring object detection to catastrophic overload or blindness." IJCAI 2024.

**Questions:**

The major challenge of using benign features as triggers is that clean training datasets might already contain these features (such as books), leading to conflicts between the trigger pattern and benign features, potentially hindering backdoor learning. I am very interested to know how the authors have approached and tried to resolve this issue.

Code implementation: I would like to see the code made open source

In conclusion, I think using natural objects as triggers is a good idea, but the challenges lie in addressing the potential conflict between benign features and the trigger, which could cause the backdoor training to fail (e.g., a significant drop in clean ACC).

**Details Of Ethics Concerns:**

I noticed the public comments and have flagged the paper for an Ethics Review, requiring further verification by the AC.

---

> ### Author Response · Authors · 2024-11-24
> **Part 1**
>
> **Q1: Typo**
>
> **A1:** Thank you for the correction, we will modify it accordingly in the final version.
>
> **Q2: Multi-label settings**
>
> **A2:** Thank you for the comment. We’d like to clarify that the statement “Wenger et al. is limited to multi-label setting” is not intended to underestimate the contributions of the work;  we completely agree with the reviewer that the key ideas in Wenger et al. can be extended to the multi-class setting as well. Here, we refer to the fact that backdoor research in the physical domain is stagnant due to the difficulty of manually creating and sharing physical datasets; for example, in order to validate the ideas of Wenger et al. in the multi-class setting, one must create a physical dataset, which is an arduous task as discussed in our paper. Motivated by these challenges, our framework solves this problem, i.e., crafting the datasets semi-automatically with powerful diffusion/generative models, to accelerate this research domain.
>
> **Q3: Challenges of using natural objects as backdoor triggers in object detection tasks versus classification tasks**
>
> **A3:** Object detection tasks are composed of 2 subtasks, which are object localization and object classification tasks. To elaborate, classification tasks are a subtask of object detection, and the ultimate goal of a backdoor trigger (natural objects in this case) is to fool/mislead the classifier into predicting a predefined target class. On the other hand, the usage of natural objects as backdoor triggers is relatively more flexible in the case of object detection tasks, which could be to mislead the object localization module, object classification module, or both. Such flexibility provides more opportunities for attackers to exploit, as discussed in Zhang et al, where SPONGE attacks the inference time of the object localization module while BLINDING attacks the accuracy of the object classification module.
>
> We use the classification task as our main task of interest to analyze the framework as we specifically want to limit such flexibility so that our analysis is feasible (e.g., the creation of physical validation dataset is less arduous). Note that, it does not mean our work is less general, as classification is still a broad, well-studied and important task in backdoor research. Extending our framework to other setting could be an interesting future direction of our work.
>
> **Q4: Is the work novel because the proposed methods are quite straightforward?**
>
> **A4:** We would like to emphasize that we do not claim that the models used in our framework are our novelty; our novelty, instead, lies in the design of the proposed framework to ease researchers of studying both physical backdoor attacks and defenses, and the rigorous analysis of this framework in this context, similar to the work in Wang et al, 2023 (VSSC), which also use a combination of straightforward generative models to achieve highly effective physical attack that is robust against physical distortions. Consequently, for backdoor research, we believe this is a significant contribution as it will help accelerate the existing physical backdoor research.
>
> **Q5: Comparison to Zhang et al.**
>
> **A5:** Zhang et al propose an attack on object detection task where the attacker is assumed to control the training process to find optimal insertion of the (potentially physical) trigger object. Consequently, the work of Zhang et al has a similar objective (i.e., to insert the trigger) as that of our Poison Generation module. In this sense, we believe that the work of Zhang et al. is orthogonal to ours, and the backdoor researcher, who requires the creation of a dataset that induces a better attack success rate, could utilize the proposed algorithm of Zhang et al to better insert the backdoor trigger suggested by our Trigger Suggestion module. Note that, our Poison Generation module also provides a specification for synthesizing samples with the trigger, besides inserting the trigger into the existing samples, when the backdoor researcher only has access to the labels (discussed in Section 4.2), which demonstrates the generality/practicality of our Poison Generation module.
>
> Finally, as discussed in our paper, we aim to enable backdoor researchers to overcome the difficulties (time-consuming and potentially difficult IRB approval effort) encountered in physical backdoor research as reported in previous studies. Specifically, the proposed framework allows backdoor researchers to conduct physical backdoor research completely from the lab and our contributions are the rigorous and extensive analysis of this framework. For these reasons, we believe that our work can help accelerate the stagnant area of physical backdoor research, which is a significant contribution to the backdoor domain.
>
> **Reference:**
>
> [1] Finding Naturally Occurring Physical Backdoors in Image Datasets
>
> [2] WaNet

---

> > ### Author Response · Authors · 2024-11-24
> > **Part 2**
> >
> > **Q6: Potential of using benign features as the trigger.**
> >
> > **A6:** Thank you for the question. The assumption of potentially using the benign features as the trigger is indeed valid, which we’re aware of. However, as we focus on the creation of the framework, we do not explicitly approach and analyze this issue, as well as many potential issues of backdoor attack such as the sizes, locations, background, lighting, patterns, reflection, wearable items, semantics, etc… of the trigger. We believe that studying these scenarios deserves several independent works based on our framework, which greatly depend on the domains and applications of the studies.
> >
> > Another important reason for omitting this issue is that it is valid for the case when the dataset is created manually or by our framework, or even when the dataset has the “digital trigger”; consequently, we believe that the specific behaviors of this scenario are not unique to physical backdoor.
> >
> > **Q7: Code release.**
> >
> > **A7:** We intend to release the code, as well as the benchmark dataset (which already received IRB approval), of our work upon the acceptance of the paper.

---

> ### Comment · Reviewer_qzqr · 2024-11-25
>
> Regarding Q2: Since the author has clarified that the approach is similar, I would suggest revising the relevant expressions in the manuscript to better reflect this.
>
> Regarding Q3: What I intended to highlight is that the idea aligns with the approach of diffusion models. At the very least, the author should include these works in the discussion.
>
> Regarding Q4: The author could include a statement in the manuscript emphasizing that their method is simple and effective. Additionally, beyond my concerns about the methodology, I believe the experimental validation is insufficient and needs further enhancement.
>
> Lastly, I noticed the public comments and have flagged the paper for an Ethics Review, requiring further verification by the AC.

---

### Official Review · Reviewer_4PYJ · 2024-10-27

**Soundness:** 2
**Presentation:** 3
**Contribution:** 2
**Rating:** 5
**Confidence:** 4

**Summary:**

This paper proposes a framework to generate poisoned backdoor datasets, which consists of three components: 1) trigger suggestion, 2) trigger generation, and 3) poison selection. The motivation is to provide a more practical, generalized, and automated framework. The advantage is that this paper takes into account physical images captured from various devices.

**Strengths:**

- The built dataset contains physical data captured by various devices.
- The framework automatically chooses the most suitable trigger, which is usually not considered by previous works.
- It is easy to follow the presentation of the paper.

**Weaknesses:**

- Lower ASR even compared to clean label attacks, such as LC [1] and Narcissus [2]. The authors need to explain if there are any challenges behind the lower ASR.
- Evaluated by very old defenses. The attack in this paper should also consider recent defenses, such as BTI-DBF [3] and IBD-PSC [4].
- Only one dataset (5 classes is very small) and one small architecture. Considering larger datasets, such as a subset of ImageNet with 100 classes but fewer samples in each class.
- According to Figure 1, the VQA model is a part of the "trigger suggestion" component, so it is not an individual contribution.
- The motivation is not clear to me. For example, in section 3, the authors mention the previous method only works in multi-label settings, but the experiments in this paper are also conducted on a multi-label (5-class) dataset. It looks like this paper does not solve the problem raised. It would be better if the authors could clarify how their framework addresses the limitations of previous methods.

[1] Label-Consistent Backdoor Attacks

[2] Narcissus: A Practical Clean-Label Backdoor Attack with Limited Information

[3] Towards Reliable and Efficient Backdoor Trigger Inversion via Decoupling Benign Features.

[4] IBD-PSC: Input-level Backdoor Detection via Parameter-oriented Scaling Consistency

**Questions:**

- Are there any experiments conducted on physical devices? As far as I can see, the authors use the devices in Appendix B to only build the dataset. Is it possible to also apply the trained model (by the poisoned dataset) to a physical device for the classification task?

- As the paper offers a toolkit for backdoor studies, do the authors consider open-source their code?

- The authors aim to provide an "effortless" framework. Are there any results about time consumption?



typo: "thsi" (line 104)

---

> ### Author Response · Authors · 2024-11-24
> **Part 1**
>
> **Q1: Lower ASR compared to clean label attacks and explanation behind lower ASR**
>
> **A1:** Our work aims to allow backdoor researchers to overcome the difficulties (time-consuming and potentially difficult IRB approval effort) encountered in physical backdoor research as reported in previous studies [1], [4]. Consequently, please note that we do not intend to achieve the best possible attacks or compare them with existing attacks such as LC or Narcissus. Rather, we focus on creating a framework that can replicate the studies in [1], [3], [4] for backdoor researchers, completely from the lab.
>
> Nevertheless, we’d like to provide a potential explanation for your question. The ASR could be affected by 3 factors, size, position, and semantics of triggers (content) as elaborated in [1]. In our case, the size of triggers is generally visible and the position of triggers is generally close to the subject matter. However, there exists a difference between the semantics of the “synthesized” physical triggers and the “real” physical triggers. As shown in [2], generally the transfer learning performance of models trained with synthetic data is lower than models trained with real data, which is mainly due to the limited diversity of the generative models. Additionally, the “synthesized” physical triggers might not be coherent with the “real” physical triggers in terms of appearance, specifically for generic types of triggers. For example, “tennis ball” would be highly probable to be consistent across the synthesized and real images, but “books” could have a high variance of representations within the synthesized images, and pose difficulties in collecting the exact similar representation. Therefore, as depicted in Tab 1-2 (main paper), the ASR of books is generally lower than that of tennis balls as physical triggers. Note that, these are similar observations as in the previous studies of physical triggers [1].
>
> **Q2: Evaluation on recent defenses**
>
> **A2:** First, we’d like to remind that our work focuses on creating a framework that can replicate the studies in [1,3,4] for backdoor researchers, completely from the lab. In the paper, we have already evaluated our framework with representative backdoor defenses, including backdoor detection and backdoor mitigation methods. Nevertheless, we followed the reviewer’s suggestion and performed the experiments with IBD-PSC [5] accordingly, with the results presented in the following table.
>
> |Trigger|TPR|FPR|AUC|F1|
> |--- |---|---|---|---|
> |Tennis Ball (Image Editing)|70.38 |9.75 |0.88|0.75|
> |Book (Image Editing)|40.88|13.55|0.73|0.52|
> |Tennis Ball (Image Generation)|99.55|20.67|0.92|0.89|
> |Book (Image Generation)|0.31|21.33|0.33|0.01|
>
> As shown in the table above, we observe that for “tennis ball”, IBD-PSC is able to effectively detect the physical trigger. We conjecture such observation is due to the “unique” characteristic of a tennis ball, where the object itself is self-explanatory and shares a common perception. These “unique” triggers would share high similarity across the synthesized images and the real images. Such a characteristic would lead to a stronger trigger strength, hence a higher Real ASR as depicted in both Tables 1 and 2 in the main paper. For “book”, which is a generic trigger, and different people might not share a common perception of it, usually would be less likely to be exactly the same across the synthesized and real images. Hence, it acts as a more “generic” trigger, which makes it harder for the model to overfit onto it, inducing lower Real ASR as depicted in both Tables 1 and 2. Since unique triggers possess a stronger trigger strength, it is easier to be detected/mitigated by existing backdoor defenses, whereas generic triggers (book) have lower trigger strength, hence we observe a lower detection rate (F1 and AUC in the above table) compared to the unique trigger (tennis ball).
>
> Nonetheless, backdoor defenses in the physical backdoor research remain challenging due to dataset collection and require extensive I/ERB approvals, our framework allows researchers to study other potential backdoor defense mechanisms in the physical domain, thus accelerating the progression of physical backdoor research. As discussed above, the result of the backdoor defense might vary significantly due to different triggers that are being injected into the dataset; thus it’s inconclusive to claim that a defense is successful against physical backdoor attacks with only one or few physical triggers. To study and prove the efficacy of physical backdoor defenses across these plethora of physical triggers requires an arduous manual effort to collect the dataset, where each dataset should be collected repeatedly with different physical triggers. Our framework is motivated to resolve such difficulties and provide researchers with a toolkit to synthesize surreal physical backdoor dataset, in order to evaluate attack/defense methods extensively and prove their effectiveness.

---

> > ### Author Response · Authors · 2024-11-24
> > **Part 2**
> >
> > **Q3: Only one dataset (5 classes is very small) and one small architecture. Considering larger datasets, such as a subset of ImageNet with 100 classes but fewer samples in each class.**
> >
> > **A3:** As we focus on creating a framework that can replicate the studies in [1,3] for backdoor researchers, we conduct similar experiments with similar numbers of classes to [1,3]. Note that this choice also makes it feasible to collect physical data to benchmark the framework. We also use the ImageNet dataset, as it is one of the most natural, complex, and most importantly publicly available datasets; the naturalness allows us to “capture” similar images in the real world for our benchmark.
> >
> > Nevertheless, we follow your suggestion and perform the experiment on a larger subset of ImageNet, with 50 classes (without capturing more real data due to the time constraint) and on additional model architectures (DeiT-Tiny and ResMLP). The experiment setting is the same as written in Section 5.1 and we have selected the “book” as the trigger with Image Editing models. We have selected 10% as the poisoning rate and the results are tabulated as follows:
> >
> > |Model|CA|ASR|
> > |-|-|-|
> > |ResNet-18|87.89|84.94|
> > |DeiT-Tiny|92.12|92.25|
> > |ResMLP|92.92|90.50|
> >
> > As observed, the CA and ASR are similar across different model architectures to that of ImageNet-5 (Table 2, main paper), even with 10x more classes.
> >
> > **Q4: According to Figure 1, the VQA model is a part of the "trigger suggestion" component, so it is not an individual contribution.**
> >
> > **A4:** We’d like to note that the goal of our paper is to introduce a framework to synthesize datasets for physical backdoor research. Choosing a suitable trigger is challenging, and demands human knowledge or a significant workload to identify a compatible trigger. In our framework, we present our Trigger Suggestion module as one of the main contributions, which aims to reduce the human cognitive effort in coming up with potential physical triggers from scratch by recommending a list of physical triggers for the user to select (as discussed in Section 3 - Motivation).
> >
> > However, the proposal of VQA as a component in the Trigger Suggestion module is not straightforward. As discussed in Section 4.1, identifying the compatible physical trigger is a non-trivial task, as it has to align with the image context, and such a task requires arduous effort from humans. Hence, we propose to leverage the general vision understanding capabilities in VQA to recommend a list of suitable objects for each image and compute the frequency of each trigger across the dataset, in order to identify the physical trigger with “moderate” compatibility.
> >
> > Additionally, we showed that the suggestions from the VQA module are highly similar to human suggestions, as depicted in Table 6, Appendix D. This highlights the importance and significance of our proposal of VQA as a component in the Trigger Suggestion module, which could reduce human cognitive effort while achieving high agreement with humans.
> >
> > We will include this discussion and clarify the contribution in the camera-ready version.
> >
> > **Q5: The motivation is not clear to me. For example, in section 3, the authors mention the previous method only works in multi-label settings, but the experiments in this paper are also conducted on a multi-label (5-class) dataset. It looks like this paper does not solve the problem raised. It would be better if the authors could clarify how their framework addresses the limitations of previous methods.**
> >
> > **A5:** First, we’d like to clarify that the setting in our work is multi-class, where each image belongs to only 1 of a predefined set of classes, and not multi-label, where each image can belong to multiple classes simultaneously.
> >
> > Section 3 discusses that the work of Wenger et al. is limited to multi-label settings, and research in this domain is stagnant due to the difficulty of manually creating and sharing datasets, for example, to study multi-class cases. Our framework solves this problem, i.e., crafting the datasets semi-automatically with powerful diffusion/generative models, to accelerate this research domain. We argue in this section that the proposed framework is suitable for both multi-label and multi-class settings, due to its lower barrier to creating the datasets.
> >
> > **References:**
> >
> > [1] Backdoor Attacks Against Deep Learning Systems in the Physical World
> >
> > [2] Fake It Till You Make It: Learning Transferable Representations From Synthetic ImageNet Clones
> >
> > [3] Finding Naturally Occurring Physical Backdoors in Image Datasets
> >
> > [4] Physical Backdoor Attacks to Lane Detection Systems in Autonomous Driving
> >
> > [5] IBD-PSC: Input-level Backdoor Detection via Parameter-oriented Scaling Consistency

---

> > > ### Comment · Reviewer_4PYJ · 2024-11-25
> > >
> > > Although some of my concerns have been addressed, the experiments are still not very convincing to me.
> > > The experiment greatly relies on a small class number, but there are more classes in the physical world.
> > > The main part of the experiment is built by using a tennis ball or book as the trigger.
> > > It is not clear to me whether the physical triggers work in a more complex setting or whether users can design specific triggers.
> > > In some cases, the ASR achieves only around 60%.
> > > A possible example to improve the experiment is using another 5 classes.
> > > Overall, the idea of using the diffusion model to generate datasets is promising but needs to be evaluated better.
> > > Given these concerns, I maintain my rating.
> > >
> > > In addition, the public comments need further verification.

---

### Official Review · Reviewer_6Ge8 · 2024-11-03

**Soundness:** 3
**Presentation:** 3
**Contribution:** 3
**Rating:** 5
**Confidence:** 4

**Summary:**

This paper proposes an automated framework for generating physical backdoor datasets using generative models to make physical backdoor attack more accesible. The framework has modules:

Trigger Suggestion - Uses VQA models to suggest suitable physical triggers
Trigger Generation -  Create poisoned samples by generating or editing
Poison Selection - Use ImageReward to select the best poisoned images

**Strengths:**

Overall, this is good work that makes physical backdoor attack research more accessible by providing a framework to generate datasets, which is usually the most tedious part.

The three-phase design makes good sense, and the results are considered comprehensive, as many aspects have been discussed, such as the common accuracy on clean inputs, attack success rate, as well as resilience, saliency heatmap, and dataset entropy.

**Weaknesses:**

My major concern is that the design of the framework appears to be a straightforward combination of a few existing solutions, i.e., a pretrained VQA model for trigger suggestion, stable diffusion or instruct diffusion for trigger generation/editing, and ImageReward for final poisoned data selection. Additionally, there is limited to no further customization or modification of these existing works to make them more integrated or collaborative. Therefore, the innovative contribution of this paper is very limited.

It also seems that we have limited control over the framework in terms of poisoned data generation. For some complicated datasets, it may be difficult to precisely control the size, type, or position of the trigger. This functionality is critical for certain tasks, given the diverse settings in the physical world. It would be beneficial to discuss this aspect in the paper and potentially incorporate it into the framework design.

As a physical backdoor dataset generator, it is more important to compare the quality of the generated images to real images, poisoned images from other related works (semantic trigger backdoor attacks have been around for quite some time), and edited images using traditional methods such as Photoshop. MORE evidence and examples need to be provided, especially in the appendix.

**Questions:**

Could you carefully justify the technical contribution of this framework?
Please discuss the customizability of the framework as pointed out in the weaknesses of the paper.
Please add more examples and comparisons of the generated poisoned images.

---

> ### Author Response · Authors · 2024-11-24
>
> Thank you for the constructive comments.
>
> **Q1: Limited innovative contribution**
>
> **A1:** Our goal is to enable researchers to study physical backdoors research effortlessly, which is currently critical, yet under-explored. Therefore, we innovatively designed this framework, which eases the research journey of physical backdoor researchers. Our framework requires minimal modification to adapt to the state-of-the-art of VQA models, generative models, and evaluation metrics. We note that our novelty lies within the combination of each of these frameworks, and proving such a combination would work in a practical scenario, by collecting and evaluating on a real-world physical backdoor dataset. Such a framework might look straightforward theoretically, but without such extensive proofs (which we have done), its practicality could not be effectively justified. Consequently, our work is especially important in accelerating physical backdoor research.
>
> **Q2: Limited control over poison data generation**
>
> **A2:** We thank the reviewer for the constructive suggestion. Our framework has incorporated 2 families of generic generative models, namely image editing (conditioned on both image and text prompts) and image generation models (conditioned on text prompts only). Both of these families cover a plethora of generative models, which our framework could leverage. Currently, our framework enables controllability over the “type” of triggers, which could be suggested by our Trigger Suggestion module or by human annotators. Controllability over the “size” and “position” could be easily achieved through our framework by adding questions to our Trigger Suggestion module such as “Where should the trigger be placed?” and “What should be the trigger size?”, and selecting Image Editing models in our Trigger Generation module to synthesize such an image. For further controllability over the Trigger Generation module, one could even utilize masked editing models (such as Blended Latent Diffusion [1]), which are conditioned on a mask, image, and text prompts, in order to synthesize a trigger accurately at the mask position. Notably, our framework requires minimal changes to achieve such controllability.
>
> It is important to emphasize that our work does not attempt to cover all possible scenarios of creating the triggers but rather focuses on providing a rigorous and extensive analysis of the main framework on variances of the ImageNet dataset. We believe that studying these scenarios deserves several independent works based on our framework, as there are many different scenarios (e.g., sizes, locations, background, lighting, patterns, reflection, wearable items, semantics, etc…), which greatly depend on the domains and applications of the studies.
>
> **Q3: More examples**
>
> **A3:** We have included more examples (25 images for each setting) as shown in Appendix H, Figures 17-20.
>
> **Reference:**
> [1] Blended Latent Diffusion

---

### Official Review · Reviewer_FLC9 · 2024-11-05

**Soundness:** 3
**Presentation:** 3
**Contribution:** 3
**Rating:** 5
**Confidence:** 2

**Summary:**

The paper presents a framework that can synthesize physical backdoor datasets. The framework consists of three modules: a trigger suggestion module that recommends suitable physical objects as triggers, a trigger generation module that creates or edits images to contain these triggers using advanced generative models, and a poison selection module that filters for the most natural-looking results. The paper demonstrates that the framework can produce datasets that achieve high attack success rates in real-world scenarios while maintaining similar properties to manually collected physical backdoor attack datasets.

**Strengths:**

- Creating physical backdoor datasets is an interesting topic and can make contributions to related work.
- Detailed discussion on improvements over existing techniques.

**Weaknesses:**

- The paper does not explain how the models in each module are trained. Also, the inputs and outputs of each step are not clear. As I understand it, in step 1 (trigger selection), the final output is a trigger. The model in step 2 then tries to attach the trigger to an image to generate the Trojan dataset. However, it seems that triggers need to be specified when training the model. So, how can this model be generalized to different triggers? Also, it is not clear what the training data is.

- From the experimental results so far, it is difficult to evaluate the realism of the generated dataset. It might be helpful to provide some generated examples.

- When specifying triggers, let's say “a car”. trigger generation may generate different cars based on its own understanding. Discussing how to ensure consistency of triggers and minimize the impact on relevant benign samples could be helpful.

**Questions:**

N/A

---

> ### Author Response · Authors · 2024-11-23
>
> Thank you for all the constructive comments and for affirming our contributions.
>
> **Q1: Explanation of each module is trained**
>
> **A1:** We clarify that we do not explicitly train each module for backdoor purposes. Within our framework, each of the modules leverages off-the-shelves models/methods. In the Trigger Suggestion Module, the output is indeed a trigger which would be then used as the text prompts for both Text-to-Image Generation and Text-to-Image Editing models, in order to inject/synthesize the trigger onto the images (at inference time). We note that there is no training involved for all three proposed modules.
>
>
> **Q2: Evaluation of the realism of the samples**
>
> **A2:** We leverage ImageReward, which is a metric for measuring human preference against synthesized images to evaluate the realism of the samples. As shown in Tables 1 and 3 of ImageReward [1], the authors have studied the correlation between human preference towards the metric and showed that ImageReward is highly correlated to human preference, thus acting as an appropriate metric to evaluate the realism of the images (since human would naturally prefer “real” images to the “fake” ones). Generally, the higher the ImageReward scores, the more a human would prefer a synthesized image over another, which quantifies the “realism” of the images. We have included some samples of generated images in Figures 2 & 8, and we will include more examples.
>
> **Q3: Consistency of the trigger**
>
> **A3:** We thank the reviewer for the constructive and valuable suggestion. First, we’d like to discuss the consistency of the trigger. Trigger objects can be broken down into 2 generic categories: unique and generic triggers. Unique triggers are self-explanatory objects, where no additional adjectives are required to describe such an object, and everyone would have the same perception of the object, given the name (tennis ball in our work); while generic triggers are objects that, if not described with adjectives, different persons would have different imagination on the objects (books in our work, or possibly cars in your example if cars are suitable triggers - recommended by VQA - for some datasets). To prevent an uncontrolled generation from the Trigger Generation module, the backdoor researcher utilizing our framework for the physical backdoor study could either use a unique trigger or add specific descriptions to the generic triggers such that to convert it to a narrower imagination space.
>
> As demonstrated in Tables 1 and 2, the tennis ball trigger (more consistent) yields a higher strength backdoor (i.e., with higher ASR) than the book trigger (less consistent). On the other hand, the trigger does not yield a significant impact on the benign samples, as we do not observe any consistent differences in clean accuracy in these experiments. We believe this is expected as the more consistent the trigger, the easier it is for the model to overfit to the backdoor, resulting in a more successful ASR, but the variation of the object should have a small (or trivial) impact on benign samples unless the triggers are common objects in the benign samples. Nevertheless, due to the flexibility of our framework, if the suggested trigger causes any interference with the benign samples (observed through the clean accuracy) and this is not the intention of the backdoor researcher, we suggest that the researcher should rely on a more consistent trigger object recommended by the Trigger Suggestion module.
>
> We will include this discussion in the final revision.
>
> **Reference:**
>
> [1] ImageReward: Learning and Evaluating Human Preferences for Text-to-Image Generation

---

> > ### Comment · Reviewer_FLC9 · 2024-11-25
> > **Response to rebuttal**
> >
> > Thanks for the rebuttal. I still have some concerns about the consistency between the description and trigger. A more formal definition about the description and some real cases could help.

---

### Public Comment · ~Ruotong_Wang2 · 2024-11-12
**Report on plagiarism**

This paper plagiarized our work presented on arXiv , “Robust Backdoor Attack with Visible, Semantic, Sample-Specific, and Compatible Triggers” (v1 and v2).
We discovered the authors' plagiarism in December 2023. Their paper published on arXiv on Dec6, 2023 [1] is highly identical to the version 1 [3] and 2 [4] of our paper.

Similarities exist in almost all parts of the proposed method:

1. **Framework** Structure: Both papers employ an identical framework comprising ‘trigger selection,’ ‘trigger generation,’ and ‘quality assessment and regeneration.’ The figure of framework in both manuscripts are remarkably alike.

2. Use of LLM for **Trigger Selection**: In their initial version [1], they used LLaVA and GPT-4 for trigger selection, similar to our method. (Their ICLR submission only mentions LLaVA this time, maybe it's to create some differences with us.)

3. Diffusion-based **Trigger Generation**: Our paper utilizes a diffusion-based text-guided image editing method for trigger insertion and mentions that this module's efficacy is expected to improve with advancements in image editing technology. Their paper also mentioned this approach and purpose.

4. **Quality Assessment and Regeneration**: We introduced quality assessment of generated images, with a procedure for regenerating images that do not meet standards. Their paper employs an identical process.

In our papers, we emphasized that each module is designed with flexibility, allowing replacement by cutting-edge technologies. Substantially, their methods are exceedingly similar to ours and have many similarities in writing as well.

It is worth noting that two authors of this paper admitted to being the Area Chair and Reviewer for our submission to NeurIPS 2023, respectively. So, it's impossible that they haven't seen our paper.

Upon uncovering these similarities, we engaged in nearly 50 days of communication with the authors of [1], both face-to-face and via email. They admitted that they began discussing and implementing our framework shortly after seeing our submission in the review stage. Finally, they withdrew their submission from CVPR 2024 and cited our paper in [2]. Because of the extensive similarity, they had to mention our work 15 times in [2].

However, in their current ICLR Submission13749, they have removed all the discussions related to our work but retained all the plagiarized sections and didn't mention any "inspiration" from our paper.

For a clearer illustration, we have listed the related evidence in the attachment, including a timeline, some similar parts, the comparison of two framework figures, and content about our paper that are deleted in this submission version. You can also compare the 2nd version of our paper [4] with Submission13749 to find the similarity, and compare Submission13749 with their arXiv version [2] to find the removed parts.

[1] Yang, S.J., La, C.D., Nguyen, Q.H., Bagdasaryan, E., Wong, K.S., Tran, A.T., Chan, C.S. and Doan, K.D., Synthesizing Physical Backdoor Datasets: An Automated Framework Leveraging Deep Generative Models. arXiv preprint arXiv: 2312.03419v1, 2023.

[2] Yang, S.J., La, C.D., Nguyen, Q.H., Bagdasaryan, E., Wong, K.S., Tran, A.T., Chan, C.S. and Doan, K.D., Synthesizing Physical Backdoor Datasets: An Automated Framework Leveraging Deep Generative Models. arXiv preprint arXiv: 2312.03419v3, 2023.

[3] Wang, R., Chen, H., Zhu, Z., Liu, L., Zhang, Y., Fan, Y. and Wu, B., Robust backdoor attack with visible, semantic, sample-specific, and compatible triggers. arXiv preprint arXiv:2306.00816v1, 2023.

[4] Wang, R., Chen, H., Zhu, Z., Liu, L., Zhang, Y., Fan, Y. and Wu, B., Robust backdoor attack with visible, semantic, sample-specific, and compatible triggers. arXiv preprint arXiv:2306.00816v2, 2023.

Supporting materials: https://drive.google.com/file/d/1JfMx36zTRc82KXswh4Ww1zltiRUWsajO/view?usp=sharing

---

> ### Author Response · Authors · 2024-11-20
>
> I (Khoa) am disappointed by this factually incorrect message from Ruotong, falsely accusing us of plagiarism, especially after the fact that we have responded to his team, with extensive evidence that our paper naturally evolved independently with a focus on another important but different topic in Backdoor Research. I intend to (1) provide all public readers with all the discussions and supporting evidence we had provided Ruotong’s team earlier to support the originality of our work, then (2), based on these records, expose many “false” claims and accusations in Ruotong’s message.
>
> To avoid burdening the readers with excessive details (if they’re not interested), I will provide 3 different versions of our public response: (A) the Short Summary, (B) the Expanded Version (with major events and evidence), and (C) the Detailed Version (with every timestamped events and communication between our team and his team). I will post only the short version (A) in this comment, while everything will be presented in this [59-page document](https://docs.google.com/document/d/1GWqcUEfeL6g4SWwLALItbDrPDyZK2zIMWL22-69oMYk/edit?usp=sharing). I encourage the readers to view version (B) as it contains better but not too overwhelming descriptions of all events.
>
> After reading one of these, or all, versions, we encourage the readers to decide for themselves whether we have conducted plagiarism of their work and whether we owe the rights to all contributions in our ICLR’s version (i.e., whether this submission has any ethical violation). If most readers think we have done so, I WILL PUBLICALLY apologize to Ruotong and his team, and WILL IMMEDIATELY REMOVE every single version of our paper online.
>
> I HAVE ZERO TOLERANCE for plagiarism and any unethical doings in my group. Life is short and research is supposed to be fun and self-fulfilling; plagiarism is definitely NOT any of those.
>
> Note that, as Submission Anonymity is now broken due to the public comment, I will refer directly to several names, some of which are the names of the authors of the paper.
>
> VSSC: The paper “Versatile Backdoor Attack with Visible, Semantic, Sample-Specific, and Compatible Triggers” from Ruotong Wang, Hongrui Chen, Zihao Zhu, Li Liu, Baoyuan Wu, with 4 versions.

---

> > ### Author Response · Authors · 2024-11-20
> > **(A) The Short Summary**
> >
> > - [Motivations] We started discussing the problem of using generative models to help advance physical backdoor research very early in 2023, focusing on creating a framework for backdoor researchers to conduct physical backdoor research. VSSC [v1], instead,  focused on creating an attack where the triggers were resistant to physical distortions.
> > - On June 30th, 2023, we had the evidence of the original development of our Trigger Suggestion and VQA, many days before NeurIPS’s reviewing deadline. Note that, however, we didn’t locate this evidence until Jan 24th, 2024.
> > - On Sep 26th, 2023, Jason presented the use of ImageReward. This is evidence of the originality of our Poison Selection module.
> > - VSSC [v2]’s released on Oct 8th, 2023, which VSSC’s authors believed we plagarize from.
> > - On Dec 8th, 2023, Hongrui Chen asked us to clarify the similarities in “structure”, “use of LLM”, “poison selection” to [v2].
> > - We explained on Dec 11th, 2023 that our focused problems are different, with evidence that we already worked on our paper with major results long before [v2].
> > - Our team met with VSSC’s team at NeurIPS where Prof. Baoyuan questioned us why we changed from “selecting trigger” to “Trigger Suggestion” and suspected us of using the NeurIPS reviews for our paper.
> >   - We provided explanation but without evidence of the original development of Trigger Suggestion.
> >   - Note that, I also told everyone that I remembered instructing Jason to use VQA to select triggers in June 2023.
> >   - We presented proof of the original development of Poison Selection (ImageReward).
> > - From NeurIPS to Jan 21st, 2024, Hongrui and Ruotong started to accuse us of “replicating their ideas” and “misleading readers”, and demanded us to write exactly what they wanted us to write in our paper (aka, every contribution we had was inspired or just extensions of some contributions in their work). We also provided our revisions during this time.
> > - These exchanged emails triggered us to defend our work more rigorously because our paper would have had close to zero novelties based on what they wanted us to write and we knew exactly that we developed them independently. Specifically, Prof. Chan explained our originalities of ImageReward/Poison Selection and Trigger Suggestion. For Trigger Suggestion, we presented the evidence of its original development before NeurIPS’ reviews, indicating that we had no motivation to use the NeurIPS reviews (of course, my team didn’t read them).
> > - Then on Jan 24th, 2024, to put the matter behind us peacefully, we revised our paper to mention that our Trigger Suggestion was inspired by their “trigger selection”, our Poison Selection was original and concurrently developed. Note that, this didn’t mean we agreed to “underestimate” our contribution/novelty as we have extensive evidence to support all our original contributions.
> > - On Oct 1st, 2024, we submitted our revised paper to ICLR. We cited VSSC but decided to remove several parts mentioning VSSC. The reason was that we strongly believed (with supported evidence) that (1) our contributions were original, (2) our motivation/focus was different from VSSC, and (3) we had to select/revise several parts of our paper to fit the story we were telling, thus we had to prioritize what we wrote. Again, I’d like to reiterate that every part of our paper was originally developed, and we now credited VSSC [v1] as a citation in our paper. We are entitled to evolve our work and write our manuscript accordingly to highlight our original contributions in a conference submission, just as the latest VSSC [v3/v4] has evolved significantly from VSSC [v1/v2].

---

> ### Author Response · Authors · 2024-11-20
> **(A) The Short Summary - Continue...**
>
> - Ruotong publicly accused us of wrongdoings, but this public comment contained several false claims, including:
>   - the plagiarism accusation → despite the fact that we clearly clarify with them all contributions in our paper were original
>   - the accusation that Tuan-Anh and Kok-seng violated NeurIPS’ reviewing ethics → despite the fact that we clearly showed them our original developments before NeurIPS’ reviews
>   - the false claim that we “admitted” to them that we began our work shortly after seeing their NeurIPS submission → we never made such an admission (and my team didn’t read the reviews)
>   - the underestimate of our work → I completely disagreed; this thought was due to the repeated statement that we “replicated” their work, which again was proven “false” with extensive evidence.
>   - the wrong accusation that we had to cite their work 15 times, which means we plagiarized → We never copied or plagiarized, as extensively demonstrated with full evidence.
>   - the false and baseless accusation that we removed all discussion related to VSSC but “retained the plagiarized sections” → despite our effort to present them with extensive evidence of our originalities in every part of our paper. But seriously, unless we were extremely foolish, why would we keep “our plagiarized sections” (while removing things that helped “prove” us clean)???
> - We also had the opportunity to read their VSSC [v3/v4]  and surprisingly found that VSSC [v3/v4] now has several similarities to our work, including the argument of labor-intensive/time-consuming process, the explicit separation into 3 modules instead of 2 as in [v2], the use of VQA/VLM to determine the trigger compatibility, among others... Note that, I am not accusing VSSC’s authors of anything (because we know very well how it feels), but I’d like to question that, if VSSC’s authors are allowed to evolve their paper with similarly motivated designs from another paper, why couldn’t we do it too (besides, we had extensive evidence to demonstrate our originalities) -- this is a hypothetical question. I clearly see that our 2 works are targeting very different, equally important problems, and both are based on generative models; thus it is reasonable to use similar tools or designs, and of course, how we use those tools is the novelties of each work.
>
> Everything I wrote in this response is from me, and all the events were prepared by Jason/Chinh/Quang, without any involvement of my co-authors, including Prof. Chan, Eugene, Tuan-Anh, and Kok-seng, as I don’t think it’s necessary for them to be involved (they’re all extremely busy just like I am); we did take almost 2 months to exchange emails with VSSC’s authors. In conclusion, we never violated any ethical standards and our work is original. Perhaps, the coincidence of having 2 people involved in NeurIPS reviewing process of a paper while my team, who mainly contributed to this project, was carrying concurrent work created several misunderstandings and false, unnecessary accusations of our work. Nevertheless, if it comes to shove, I am sure all of my co-authors and I are ready to peacefully collaborate to defend our integrity.

---

> ### Public Comment · ~Ruotong_Wang2 · 2024-11-23
>
> I am very disappointed with the team led by such an experienced researcher like Koah. Clearly, Yang needs further education on academic ethics and integrity.
>
> During December 2023 to January 2024, we provided Koah's team ample time to prepare evidence. They failed to demonstrate the independence of their work, and had to withdraw their paper and modify the arXiv version. Surprisingly, they suddenly found more evidence after our public comment.
>
> However, the evidence Koah provided only further confirms their academic misconduct. We must once again emphasize the following facts:
>
> 1. Yang's paper published on December 6, 2023, shows striking similarities in content and figures with our paper published on October 8, 2023. This far exceeds coincidence and requires no additional proof.
> 2. Yang immediately analyzed our paper upon its release and our work was mentioned many times in their discussion. Yang even acknowledged in discussions (Sept. 12, 2023) that InstructDiffusion they planned to use, was similar to Instruct-pix2pix we used in VSSC. However, Yang avoided citing us in their initial version and refused to acknowledge our inspiration in this submission.
> 3. There is no evidence to support that Tuan-Anh and Kok-seng (AC and reviewer in NeurIPS 2023) did not communicate with Koah’s team, given that they are co-authors of the paper.
>
> Although plagiarism should be determined by the release dates and content of the papers - which is already totally obvious - we are still willing to provide a detailed analysis of the documents provided by Koah:
>
> **May 22, 2023:** Their documents show that before seeing our paper, they used a completely different method to obtain poisoned images: image generation based on DreamBooth.
>
> **June 1, 2023:** We released the first version of VSSC (v1).
>
> **June 6, 2023:** Yang conducted a detailed analysis of VSSC[v1]. This is the point where their methods started to show suspicious similarities to ours.
>
> **June 28, 2023:** Yang began using the same approach as ours, incorporating triggers into images through image editing method. At this point, based on their analysis of our paper, they manually selected trigger based on object name and appearance.
>
> **August 2023:** During rebuttal, a reviewer asked about our trigger selection, and we provided a detailed explanation on how we used GPT-4 to generate candidate lists for trigger selection.
>
> **September 7, 2023:** At this point, they were very concerned about the trigger selection method that we were asked during rebuttal period, but they focused on "Finding compatibility between a class object and a trigger object”, which was still different from ours.
>
> **September 11, 2023:**  Their first try to use our trigger selection method (which we detailed explained to reviewers during NeurIPS rebuttal) by directly querying LLM for suitable triggers. Before this, they used image captions to find co-existing objects.  It is hard not to suspect the information leakage from AC and reviewer.
>
> **September 12, 2023:** At this point, Yang hadn't determined their trigger generation method, but clearly mentioned InstructDiffusion they planned to use(also what they finally used) was similar to the method used in our VSSC: generating triggers using diffusion-based image editing.
>
> They analyzed our low-quality samples and attempted to use the same trigger as ours.
>
> **September 29, 2023:** Continued discussing on our work. (then "forgot" to cite us in their first version in December). It seems that Yang planned to differentiate their work from VSSC through image selection. However, while they claimed to “discard implausible samples” in Section 4.3 and 5.4 of their submission,  they mistakenly drew the regeneration process of VSSC in their flowchart (Figure1). This mistake is perhaps a result of excessive 'reference' to our paper. It’s really an amusing mistake that reveals some facts.
>
> **October 8, 2023:** We released v2 of VSSC, adding Quality Assessment Module and regenerating low-quality samples. In the paper Yang released on **December 6, 2023**, they used the same regeneration process as ours, whereas previously they simply discarded low-quality samples.
>
> **December 6, 2023:** Yang released the first version of their paper, with surprisingly similar flowcharts and content to our v2. This paper came six months after our VSSC[v1] and two months after our [v2]. Considering the fact that they have consistently centered their discussions around our work, it is unbelievable that Koah did not notice these striking similarities. It is hard not to suspect they deliberately avoided citing our work.
>
> All the evidence has been presented in:
> https://docs.google.com/document/d/1BVJnG_yUgobfcm57033nG3jVC1bQKHTP4spu8o3Pymo/edit?usp=sharing

---

> > ### Public Comment · ~Ruotong_Wang2 · 2024-11-23
> >
> > Finally, Koah's attempt to turn the tables by accusing our v3 is utterly absurd. As the victims who need no self-justification, we will nevertheless patiently refute these claims to end this farce:
> >
> > 1. We emphasized 9 times in our ICLR 2024 rebuttal (November, 2023) that our method addresses the labor-intensive issues in current physical backdoor attacks. How could we possibly be “similar to” their paper that wasn't released until December? The direction of similarity can only be one way: later one being similar to earlier one.
> > 2. We already used GPT-4 for trigger selection in the work submitted to NeurIPS 2023, and immediately applied it for quality assessment after the release of GPT-4v on September 26, 2023. Yang’s paper released in December used a trigger selection method similar to ours, but they insinuate that we mimicked their method.
> > 3. In our VSSC[v2] released on **October 8, 2023**, we had already proposed trigger selection, trigger insertion module, and quality assessment module. Our v3 is a direct evolution of these foundations. The fact that Yang’s first version in December contains identical components should be viewed as their similarity to our work, not the reverse. Their claim is utterly ridiculous.

---

> > > ### Author Response · Authors · 2024-11-25
> > > **My FINAL comment on this issue!**
> > >
> > > Dear Ruotong,
> > >
> > > This will be **my last message to you for a few reasons**. First, the name is Khoa (not Koah) which you repeatedly called incorrectly.
> > >
> > > I see that no matter what I do (including transparently putting every record of our work and communications with you online), you still “decided” to ignore them and conclude that we “plagiarized”, now and then. Seeing your statement
> > > > During December 2023 to January 2024, we provided Koah's team ample time to prepare evidence. They failed to demonstrate the independence of their work,
> > >
> > > I realized that you chose to ignore the facts we presented to you then. You even went too far by saying
> > > > Surprisingly, they suddenly found more evidence **after our public comment**”,
> > >
> > > while **the major events in our document are those we ALREADY sent you via emails from December 2023 to January 2024**. This question is also amusing: it implies one of us “worked” for these companies (Google, Notion) as that was the only way we could "plagiarize" those records (our understanding from your statement of “suddenly found them")? **Again, you chose to ignore whatever facts we presented, then and now, because it’s your way or the highway**!
> > >
> > > Perhaps, this stubbornness (after we’ve presented so many facts) is because you overestimated your capability while underestimating the possibility that we independently and concurrently evolved our work. **Yes, we read your VSSC [v1], which we now cited, but that’s the end of it!** I (or rather we) will choose to step away from this conversation, as we don’t want to waste your time, our time, and everyone else’s time; read the document we released if you want to find any truth or make any conclusions: it’s up to you.
> > >
> > > Since day 1 (at NeurIPS to Prof. Baoyuan) and in one of our emails to you (in the document), I already said: “I will put my academic career on the line to defend Jason (or Yang) as I knew he didn’t plagiarize!” Jason left my group at the beginning of this year, but he will always have my support on this matter; Jason is young, and he might make some mistakes but he did not plagiarize. For me (now that you accused me of plagiarism too), I simply hope you know this: committing plagiarism is a disrespect to my collaborators, my family (who sacrificed so much for me to do research), my advisors, and my decision to leave my well-paid and well-respected industry jobs for academia.
> > >
> > > My final note: I specifically said that **I did not claim you copied our work in my previous response**. I said that as your paper evolved, there are arguments and use of tools, which are becoming more and more similar to our work. I wanted to point out that it's possible for your work to naturally progress with similarities to ours as none of our papers really created new models or methods or used any proprietary technologies (*they are really straightforward*). Why we can accept that it's ok for you to have these similarities (and you had proof - we do read your previous VSSC [v1] at the beginning and [v2] after our first engagement, thus we never accused you of anything), but you cannot accept that we have similarities with yours (and we also had evidence/facts in the document)?
> > >
> > > PS1. I initially intended to write a longer response (with sub-responses to every claim you made), but all my collaborators suggested that I should make it short, because they didn’t want me to spend more time on back-and-forth discussions with you; last time it was almost 2 months.
> > >
> > > PS2. None of these events changes my respect for Prof. Baoyuan and his many important contributions to backdoor research.
> > >
> > > Cheers,\
> > > Khoa

---

### Meta-Review · Area_Chair_hkAn · 2024-12-19

**Metareview:**

The paper proposed a backdoor framework to generate malicious, physical backdoor dataset based on generative models. The framework involves 3 modules: suggesting the suitable physical triggers, generating the poisoned candidate samples, and finally refining for the most plausible ones. The experiments show the effectiveness of achieving high attack success rate on real physical world data.

The paper has been publicly questioned for plagiarism. There has been a long discussion between the paper authors and the authors of an arXiv paper. For the plagiarism issue, the AC carefully read both papers and examined their similarities in ideas, methods, and writing. Based on the assessment, it was found that while there is significant similarity in ideas between the two papers, the submission does not exhibit direct plagiarism.

However, the novelty of the paper could be limited given the prior work. There are also many concerns raised by the reviewers, such as unclear motivation of the whole framework, limited contribution, lack of experiments in some aspects. Given the consensus among all the reviewers, the AC recommends the rejection of this paper.

**Additional Comments On Reviewer Discussion:**

The reviewers have raised several concerns of the paper:

- Reviewer FLC9 raised the concerns about unclear training details and realism of the generated dataset.

- Reviewer 6Ge8 raised the concerns about limited contribution, limited control over the framework, and quality of the generated images.

- Reviewer 4PYJ raised the concerns about lower performance compared to existing methods, evaluation by old defenses, limited experiments on one dataset and one architecture, and unclear motivation.

- Reviewer qzqr raised the concerns about writing quality, limited novelty, and insufficient insights.

The authors have provided the rebuttal to address these concerns. However, the reviewers still think that this paper fall short of the acceptance bar and recommend rejection at the end. Besides, the paper has also undergone plagiarism issues by a public comment. Although AC did not think it is a direct plagiarism, there is indeed a significant similarity in ideas between the submission and an arXiv paper. Therefore, given the limited novelty and contribution, the AC would recommend rejection.

---

### Decision · Program_Chairs · 2025-01-22

Reject